# Reciprocal carbonyl–carbonyl interactions in small molecules and proteins

Abdur Rahim[1], Pinaki Saha[1], Kunal Kumar Jha[1], Nagamani Sukumar[1] & Bani Kanta Sarma[1]

Carbonyl-carbonyl $n{\rightarrow}\pi^{*}$ interactions where a lone pair ($n$) of the oxygen atom of a carbonyl group is delocalized over the $\pi^{*}$ orbital of a nearby carbonyl group have attracted a lot of attention in recent years due to their ability to affect the 3D structure of small molecules, polyesters, peptides, and proteins. In this paper, we report the discovery of a "reciprocal" carbonyl-carbonyl interaction with substantial back and forth $n{\rightarrow}\pi^{*}$ and $\pi{\rightarrow}\pi^{*}$ electron delocalization between neighboring carbonyl groups. We have carried out experimental studies, analyses of crystallographic databases and theoretical calculations to show the presence of this interaction in both small molecules and proteins. In proteins, these interactions are primarily found in polyproline II (PPII) helices. As PPII are the most abundant secondary structures in unfolded proteins, we propose that these local interactions may have implications in protein folding.

[1] Department of Chemistry, School of Natural Sciences, Shiv Nadar University, Dadri, Uttar Pradesh 201314, India. Abdur Rahim and Pinaki Saha contributed equally to this work. Correspondence and requests for materials should be addressed to B.K.S. (email: banikanta.sarma@snu.edu.in)

Nature effectively uses combinations of weak noncovalent interactions in the functional forms of various biologically important molecules such as nucleic acids and proteins[1–3]. Intermolecular noncovalent interactions of varying magnitude are also responsible for the existence of different states of matter[4]. Carbonyl-carbonyl (C=O⋯C=O) $n{\to}\pi^*$ interactions where one of the lone pairs ($n$) on the oxygen atom of a carbonyl group is delocalized over the antibonding $\pi^*$ orbital of a nearby carbonyl C=O bond ($\pi^*_{C=O}$) along the Bürgi-Dunitz trajectory[5] (∠O⋯C=O ~ 109°) have attracted a great deal of attention in recent years[6–11]. Previous studies have shown that C=O⋯C=O $n{\to}\pi^*$ interactions not only influence geometries of important small molecules[12–15] but also play crucial roles in determining the three dimensional structures of polyesters[16], peptides[17], peptoids[18–21] and proteins[22–25]. C=O⋯C=O interactions between the side-chain and backbone carbonyl groups of Asp, Asn, Glu, and Gln were also observed in the high-resolution crystal structures of proteins[26, 27]. C=O⋯C=O $n{\to}\pi^*$ interaction is characterized by a short O⋯C=O distance (d) of less than 3.22 Å [the sum of van der Waals radii of carbon and oxygen atom[28]], bond angle ∠O⋯C=O ($\theta$) of ~109° and the pyramidality ($\Delta$, $\Theta$) of the acceptor carbon atom towards the donor oxygen atom[9, 14, 17, 25, 29]. Direct spectroscopic evidence for $n{\to}\pi^*$ interaction was recently reported by using gas-phase infrared spectroscopy[30].

We anticipated that due to $n{\to}\pi^*$ interaction both donor and acceptor C=O bonds will be polarized, which will make the acceptor carbonyl oxygen atom a better electron donor and the donor carbonyl carbon atom a better electron acceptor. The acceptor carbonyl oxygen, therefore, can donate electrons to another nearby carbonyl carbon either to form a sequential chain of O⋯C contacts (Fig. 1a) or it can donate electrons back to the original donor carbonyl carbon atom forming "reciprocal" $n{\to}\pi^*$ interactions (Fig. 1b). Although, the sequential $n{\to}\pi^*$ interactions were previously observed in poly(lactic acid)[16] and proteins[22], reciprocal $n{\to}\pi^*$ interactions remained unexplored. Allen and coworkers reported anti-parallel arrangements of carbonyl groups in ketone dimers that were bound together by two intermolecular C=O⋯C=O short contacts of dipolar nature[31]. Maccallum et al reported a similar geometrical arrangement of carbonyl groups in right-twisted β-strands and observed two chemically distinct dipolar C=O⋯C=O short contacts[32]. However, these C=O⋯C=O short contacts were considerably longer than the sum of van der Waals radii of C and O atoms.

In this paper, we hypothesized that the polarization of the carbonyl groups by $n{\to}\pi^*$ interactions should lead to back and forth donations between the carbonyl pairs. Based on our hypothesis, we discovered the presence of "reciprocal C=O⋯C=O interactions" both in small molecules and proteins. To establish the existence of reciprocal C=O⋯C=O interactions, we designed and synthesized model compounds and carried out X-ray crystallographic and theoretical studies. Further, we carried out Cambridge Structural Database (CSD)[33] and Protein Data Bank (PDB)[34] analyses to show that these interactions are widely present in small molecules and proteins. In proteins, these interactions are primarily found in random coils and turn

regions. Based on our observations we propose that reciprocal C=O⋯C=O interactions may be a key local interaction that restricts the number of conformers of unfolded proteins and may have a role in protein folding.

## Results

**Reciprocal carbonyl-carbonyl interactions in $N,N'$-diacylhydrazines.** To test our hypothesis of reciprocal $n{\to}\pi^*$ interactions, we have synthesized $N,N'$-diacylhydrazines **1**-**8** having various substituents on either side of the carbonyl groups (Fig. 2a). In $N,N'$-diacylhydrazines **1**-**8**, the two amide carbonyl groups [CO-I and CO-II; Fig. 2b] are separated by three covalent bonds and 1,5-type $n{\to}\pi^*$ interactions are feasible from both sides. We propose that due to the repulsion between the nitrogen lone pairs, the $N,N'$-diacylhydrazines should be nonplanar with the carbonyl groups orientated favorably for reciprocal $n{\to}\pi^*$ interactions. Incorporation of electron donating and withdrawing substituents near the carbonyl groups in **1**–**8** should help us to tune these interactions.

As anticipated, the $N,N'$-diacylhydrazines (**1**-**8**) crystallized in nonplanar form with the carbonyl groups oriented almost orthogonal to each other (C=O⋯C=O dihedral angle = +70° to +85° or −70° to −85°) (Supplementary Table 1 and Supplementary Fig. 1). We observed that in compounds **1**–**3**, that lack any strong electron donating or withdrawing substituent near the carbonyl groups (i.e., $X = CH_3$, $CH_3CH_2$; $Y = H$, $CH_3$), the two carbonyl groups stay far apart. The crystallographic distances $d_1$ and $d_2$ (Fig. 2b) in **1**–**3** are longer than 3.22 Å (Table 1) and natural bond orbital (NBO)[35] calculations carried out on the high-resolution crystal geometries of **1**–**3** (Supplementary Table 2) show no evidence of $n{\to}\pi^*$ interaction (Table 1 and Supplementary Table 3). In compound **4**, when $X$ is an electron withdrawing group (e.g., $X = CH_2Cl$), an increase in the acceptor ability of the carbonyl CO-II is expected, which should increase $n{\to}\pi^*$ interaction from the oxygen atom of CO-I to the $\pi^*_{C=O}$ orbital of CO-II. However, the inductive electron withdrawal of Cl can be negated by the electron donation from the Cl lone pairs into the antibonding orbitals ($\sigma^*$ and $\pi^*$) of the adjacent carbonyl group (CO-II). In compound **4**, we observed such electron delocalizations from the Cl lone pairs to both $\sigma^*$ and $\pi^*$ orbitals of the C=O bonds of CO-II group, which contributed 0.67 kcal mol$^{-1}$ to the stabilization (Supplementary Table 4). Such electron donation should enhance the donor ability of CO-II in **4**. A short crystallographic O$^2$⋯C$^1$ distance (d$_2$ = 3.037 Å is shorter than the sum of van der Waals radii of C and O) and presence of NBO second order perturbation energy [$E^2_{(n{\to}\pi^*)}$ = 0.34 kcal mol$^{-1}$] for $n{\to}\pi^*$ interaction from CO-II to CO-I in **4** supports this assumption (Table 1). We anticipate that due to this electron donation from CO-II to CO-I, the CO-I group in **4** will be polarized and the carbonyl oxygen of CO-I will become a better donor. We clearly observed back donation of electrons from CO-I to CO-II in **4** as evidenced by a short crystallographic O$^1$⋯C$^2$ distance d$_1$ of 3.103 Å and $n{\to}\pi^*$ interaction energy [$E^1_{(n{\to}\pi^*)}$] of 0.10 kcal mol$^{-1}$ obtained by NBO analysis. This is in accordance with our hypothesis that donation from CO-II to CO-I increases the donor ability of CO-I and acceptor ability of CO-II thereby inducing a back donation of electrons from CO-I to CO-II. Similarly, when $Y$ is an electron donating group (e.g., $Y = OCH_3$), CO-I is expected to be a better donor and, accordingly, we observed short distance d$_1$ in compound **5** (Table 1). We also observed back donation from CO-II to CO-I in **5** (Table 1).

Among the synthetic compounds **1**–**8**, significantly shorter d$_1$ and d$_2$ are observed in **6**–**8** where electron donating or withdrawing groups are present on both sides of the carbonyl groups. The second order perturbation energies obtained by NBO

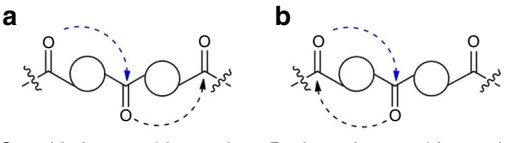

**Fig. 1** Schematic illustration of **a** one-sided and **b** reciprocal $n{\to}\pi^*$ interactions. Curved dotted arrows indicate $n{\to}\pi^*$ interactions

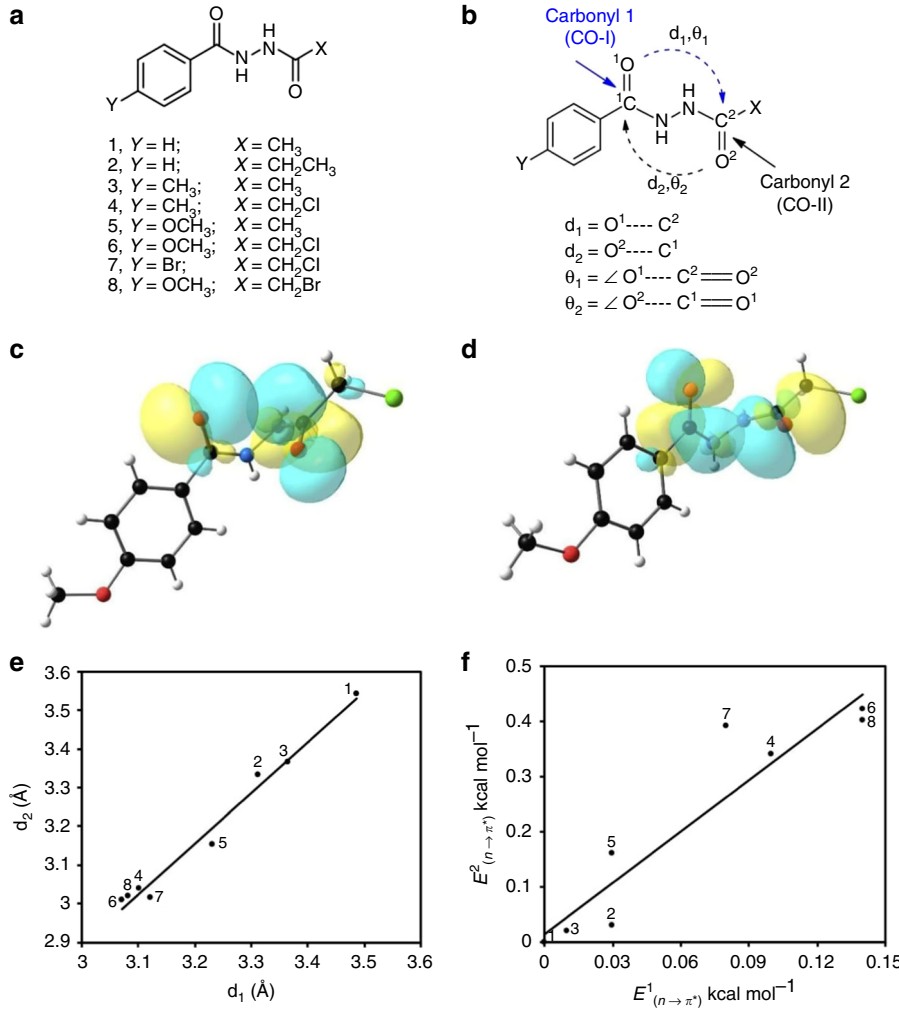

**Fig. 2** Model compounds synthesized to study reciprocal $n{\rightarrow}\pi^{\star}$ interactions. **a** Chemical structures of *N,N'*-diacylhydrazines (**1–8**). **b** Definition of different structural parameters in *N,N'*-diacylhydrazines **1–8**; $d_1 = O^1{\cdots}C^2$; $d_2 = O^2{\cdots}C^1$; $\theta_1 = \angle O^1{\cdots}C^2 = O^2$; $\theta_2 = \angle O^2{\cdots}C^1 = O^1$. **c** NBO orbital overlap between oxygen lone pair ($n_O$) of CO-I and $\pi^{\star}_{C=O}$ orbital of CO-II of compound **6**. **d** NBO orbital overlap between oxygen lone pair ($n_O$) of CO-II and $\pi^{\star}_{C=O}$ orbital of CO-I of compound **6**. **e** Plot showing correlation between O⋯C distances ($d_1$ and $d_2$) in compounds **1–8** [Linear fitting; Pearson correlation coefficient = 0.9906]. **f** Plot showing correlation between reciprocal $n{\rightarrow}\pi^{\star}$ interaction energies [$E^1_{(n{\rightarrow}\pi^{\star})}$ and $E^2_{(n{\rightarrow}\pi^{\star})}$] in compounds **1–8** [Linear fitting; Pearson correlation coefficient = 0.938]. *Curved dotted arrows* indicate $n{\rightarrow}\pi^{\star}$ interactions

calculations are also relatively higher for these compounds (Table 1). The NBO orbital overlaps between the oxygen lone pairs ($n_O$) and $\pi^{\star}_{C=O}$ orbitals in compound **6** are shown in Figs. 2c, d. Note that in compounds **4** and **6–8** where $X = CH_2Cl$ or $CH_2Br$, $d_2$ is shorter than $d_1$ and stronger $n{\rightarrow}\pi^{\star}$ interactions from CO-II to CO-I are observed by NBO analysis. This is due to the electron donation from the Cl or Br atom to the $\sigma^{\star}_{C=O}$ and $\pi^{\star}_{C=O}$ orbitals of CO-II, which increases the donor ability of the CO-II oxygen atom (Supplementary Table 4). Such electron donations from α-halogens to carbonyl groups and their effect on $n{\rightarrow}\pi^{\star}$ interactions were previously reported in the literature[36, 37].

Interestingly, the values of ∠O⋯C = O angles $\theta_1$ and $\theta_2$ are much smaller (~82°) in compounds **4–8** where reciprocal $n{\rightarrow}\pi^{\star}$ interactions are observed than in **1–3** that lack $n{\rightarrow}\pi^{\star}$ interactions (Table 1). In fact, the values of $\theta_1$ and $\theta_2$ are much smaller than what is expected for one-sided $n{\rightarrow}\pi^{\star}$ interactions (∠O⋯C = O ~ 109°) reported previously[6]. This may be due to the geometrical arrangement required for reciprocal $n{\rightarrow}\pi^{\star}$ interactions, which forces $\theta_1$ and $\theta_2$ away from the Bürgi-Dunitz trajectory.

Another important signature of $n{\rightarrow}\pi^{\star}$ interactions is the pyramidality of the acceptor carbonyl carbon atom measured by parameters Δ and Θ[9, 14, 17, 25, 29]. Positive values of Δ and Θ indicate pyramidalization of the acceptor carbonyl carbon towards the donor oxygen atom whereas negative values of Δ and Θ indicate pyramidalization of the acceptor carbon away from the donor oxygen atom. In compounds **1–8**, however, we have not observed a correlation of pyramidality (Θ) with O⋯C distance and the strength of $n{\rightarrow}\pi^{\star}$ interactions. One reason for this could be the stronger donation from the α-halogen atoms to the nearby carbonyl, which would force the acceptor carbonyl carbons towards the halogen atoms away from the donor oxygen atoms. Also, the crystal packing forces may have some influence in the observed geometries and the pyramidalization of the two nitrogen atoms between the carbonyl groups may influence the pyramidalization of the acceptor carbonyl carbons. Moreover, the individual $n{\rightarrow}\pi^{\star}$ interactions in compounds **1–8** may not be strong enough to exert a significant effect on pyramidalization of the carbonyl carbons.

Overall, these data suggest that, in compounds **1–8**, the geometrical constraints imposed by the repulsion between

**Table 1 X-ray crystallographic structural parameters and NBO data for compounds 1–8**

| Compounds | Y | X | $d_1$ | $d_2$ | $\theta_1$ | $\theta_2$ | $\Delta_1$ | $\Delta_2$ | $\Theta_1$ | $\Theta_2$ | n→π* (kcal mol$^{-1}$) | | Total n→π* (kcal mol$^{-1}$) |
|---|---|---|---|---|---|---|---|---|---|---|---|---|---|
| | | | (Å) | (Å) | (°) | (°) | (Å) | (Å) | (°) | (°) | $E^1_{(n\to\pi^*)}$ | $E^2_{(n\to\pi^*)}$ | $E^t_{(n\to\pi^*)}$ |
| 1 | H | CH$_3$ | 3.487 | 3.542 | 98.7 | 95.9 | 0.018 | 0.007 | 2.39 | 0.99 | NP | NP | NP |
| 2 | H | CH$_2$CH$_3$ | 3.314 | 3.331 | 93.7 | 93.2 | 0.013 | 0.026 | 1.70 | 3.43 | 0.03 | 0.03 | 0.06 |
| 3 | CH$_3$ | CH$_3$ | 3.367 | 3.365 | 92.8 | 92.7 | 0.008 | 0.015 | 0.99 | 1.89 | 0.01 | 0.02 | 0.03 |
| 4 | CH$_3$ | CH$_2$Cl | 3.103 | 3.037 | 82.8 | 85.6 | −0.015 | 0.000 | −1.96 | 0.012 | 0.10 | 0.34 | 0.44 |
| 5 | OCH$_3$ | CH$_3$ | 3.233 | 3.152 | 81.8 | 85.4 | 0.005 | −0.005 | 0.63 | −0.69 | 0.03 | 0.16 | 0.19 |
| 6 | OCH$_3$ | CH$_2$Cl | 3.072 | 3.009 | 82.1 | 84.9 | −0.005 | −0.019 | −0.70 | −2.59 | 0.14 | 0.42 | 0.56 |
| 7 | Br | CH$_2$Cl | 3.123 | 3.014 | 81.3 | 86.2 | −0.019 | −0.009 | −2.53 | −1.17 | 0.08 | 0.39 | 0.47 |
| 8 | OCH$_3$ | CH$_2$Br | 3.082 | 3.017 | 81.9 | 84.8 | −0.014 | −0.017 | −1.92 | −2.34 | 0.14 | 0.40 | 0.54 |

The calculations were carried out at B3LYP/6-311 + G(2d,p) level of theory. $d_1 = O^1 \cdots C^2$; $d_2 = O^2 \cdots C^1$; $\theta_1 = \angle O^1 \cdots C^2 = O^2$; $\theta_2 = \angle O^2 \cdots C^1 = O^1$ (see Fig. 2b). $E^1_{(n\to\pi^*)}$ is the NBO second order perturbation energy for electron donation from oxygen lone pair ($n_O$) of the first carbonyl group (CO-I) to the $\pi^*_{C=O}$ orbital of the second carbonyl group (CO-II). $E^2_{(n\to\pi^*)}$ is the NBO second order perturbation energy for electron donation from oxygen lone pair ($n_O$) of the second carbonyl group (CO-II) to the $\pi^*_{C=O}$ orbital of the first carbonyl group (CO-I). $E^t_{(n\to\pi^*)} = E^1_{(n\to\pi^*)} + E^2_{(n\to\pi^*)}$
NP not present

the nitrogen lone pairs orient the two carbonyl groups favorably for reciprocal n→π* interactions. We could tune these interactions by introducing electron donating or withdrawing substituents near the carbonyl groups. Interestingly, we observed that an increase in n→π* interaction from one side also leads to an increase in the n→π* interaction from the other side in compounds 1–8. This correlation suggests that n→π* interactions in these compounds could be synergistic (Figs. 2e, f). For example, shorter $d_1$ and higher $E^1_{(n\to\pi^*)}$ values are observed in 4 compared to 3 although 3 and 4 have same the substituent (4–CH$_3$–Ph) attached to CO-I. Similarly, higher donation from CO-I to CO-II is observed in 6 compared to 5 although 5 and 6 have same the substituent (4–OCH$_3$–Ph) attached to CO-I.

To find out if geometry optimization has any effect on the computed n→π* interactions in comparison to the unrelaxed X-ray geometries, we also carried out geometry optimizations in compounds 1–8 by freezing the dihedral angles of the side chains involved in reciprocal interactions to their X-ray values and freely optimizing the remaining degrees of freedom (bond lengths, angles, and dihedrals) (Supplementary Fig. 2). We observed that reciprocal n→π* interactions were retained after geometry optimizations but they became slightly weaker than what were observed from the NBO calculations on the crystal geometries (Supplementary Table 5). The coordinates of the optimized geometries of 1–8 are provided in Supplementary Data 1. We also observed that, during gas phase geometry optimization, in absence of any packing and intermolecular forces that are present in the X-ray geometries, the Cl or Br atoms attached to the methylene carbons in 4, 6–8 moved to an anti-periplanar geometry (trans) with respect to the oxygen atom of the nearby carbonyl group (CO-II). This is probably due to higher hyperconjugative delocalization between the halogen lone pairs and carbonyl π* orbital in the anti-periplanar geometry that would provide more stability to the isolated gas phase molecule. Note that such elongation of carbonyl-carbonyl (O···C) short contacts (weakening of n→π* interactions) in gas phase optimized geometry relative to the X-ray geometries are well known[9, 13, 14].

**Reciprocal carbonyl-carbonyl interactions in small organic molecules.** To probe whether intramolecular reciprocal C=O···C=O interactions are also present in other small molecules we carried out a CSD search. In our search, we looked for organic molecules having at least two carbonyl groups with intramolecular $O^2 \cdots C^5$ ($d_1$) and $O^6 \cdots C^1$ ($d_2$) distances ≤ 3.2 Å (Supplementary Fig. 3). The search was carried out for cases where the two carbonyl groups are separated by at least three

covalent bonds (1,5-type interaction). No restriction was imposed on the ∠O···C = O angles ($\theta_1$ and $\theta_2$) during the search. The CSD search provided 1432 molecules which fulfilled our search criteria (Supplementary Table 6).

The plots showing the distribution of O···C distances ($d_1$ and $d_2$) and ∠O···C = O angles ($\theta_1$ and $\theta_2$) of all the molecules obtained from the CSD search are shown in Figs. 3a, b, respectively. As can be seen from Fig. 3a, in most cases $d_1$ and $d_2$ fall in 2.90–3.20 Å range indicating that reciprocal interactions are in general weak. The values of $\theta_1$ and $\theta_2$ are mainly concentrated in the 70–100° range with majority of the molecules having $\theta_1$ and $\theta_2$ in the range 80–90°. Interestingly, we also observed similar values for O···C distances ($d_1$ and $d_2$) and ∠O···C = O angles ($\theta_1$ and $\theta_2$) in compounds 4–8 that showed reciprocal n→π* interactions. Therefore, it is quite clear that the ∠O···C = O ($\theta$) angle deviates significantly from the Bürgi-Dunitz trajectory in reciprocal C=O···C=O short contacts. The $d_1$ vs. $\theta_1$ and $d_2$ vs. $\theta_2$ plots (Figs. 3c, d, respectively) show that when the angle of approach of donor oxygen atoms to the acceptor carbonyl C=O bonds deviates from Bürgi-Dunitz trajectory, the O···C distances ($d_1$ and $d_2$) increase, suggesting weakening of interactions. NBO analyses of crystal geometries of 30 randomly chosen molecules (Supplementary Fig. 4) having $d_1$ and $d_2$ ≤ 3.20 Å and covering the range of observed ∠O···C = O angles ($\theta$) values (70–100°) showed the presence of reciprocal interactions in them (Table 2). The NBO orbital overlaps between the oxygen lone pairs ($n_O$) and $\pi^*_{C=O}$ orbitals in one such molecule (Fig. 3e) (CCDC ref. code: JUHQEK) are shown in Figs. 3f,g.

In most of the molecules obtained from the CSD search, reciprocal C=O···C=O interactions were stabilized by both n→π* and π→π* interactions between the carbonyl groups (Table 2 and Supplementary Table 7). We observed substantial C=O···C=O π→π* interactions in molecules having $\theta_1$ and $\theta_2$ values > 90° (Supplementary Table 7). In some cases, π→π* interactions are even stronger than n→π* interactions. When $\theta_1$ and $\theta_2$ values were <90°, π→π* interactions were observed for molecules having relatively shorter O···C distances (both $d_1$ and $d_2$ <2.90 Å) and stronger n→π* interactions. We propose that although the contribution of individual orbital interaction is small, the overall contribution of two n→π* and two π→π* interactions to the stabilization of molecules having reciprocal C=O···C=O interactions could be significant. Based on the NBO calculations at B3LYP/6-311 + G(2d,p) level, we observed that reciprocal C=O···C=O interactions contribute 0.11–3.37 kcal mol$^{-1}$ (with an average value of 0.98 kcal mol$^{-1}$) to the stabilization of small molecules (see the last column in Supplementary Table 7).

We observed positive values of Δ and Θ for the carbonyl carbons in most of the molecules from the CSD listed in Table 2,

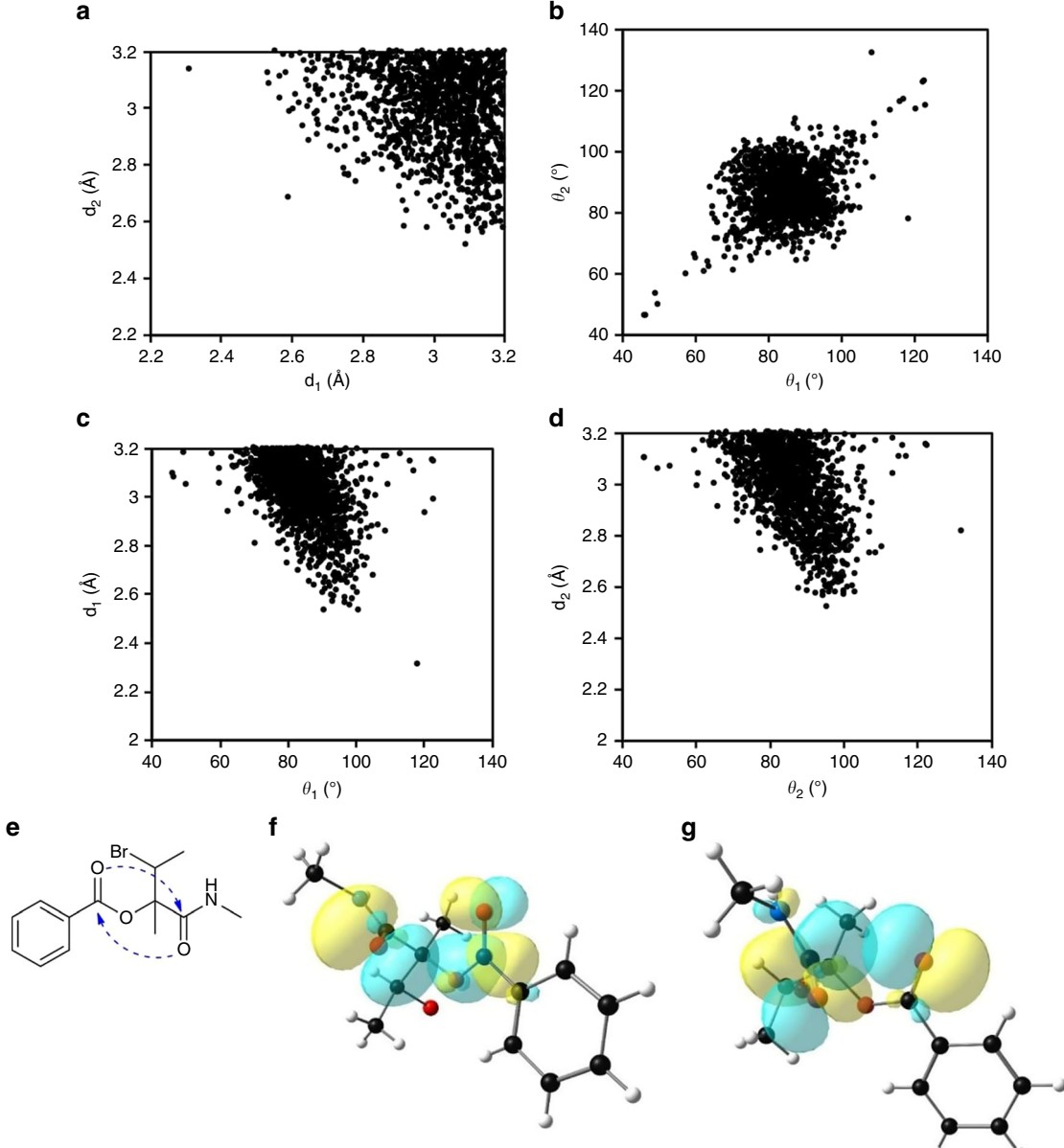

**Fig. 3** X-ray crystallographic data and NBO overlap diagrams for CSD molecules. **a** Plot showing the distribution of O···C distances ($d_1$ and $d_2$) in molecules obtained from the CSD search. **b** Plot showing the distribution of ∠O···C = O angles ($\theta_1$ and $\theta_2$) in molecules obtained from the CSD search. **c** Plot of distance $d_1$ vs. angle $\theta_1$ in molecules obtained from the CSD search. **d** Plot of distance $d_2$ vs. angle $\theta_2$ in molecules obtained from the CSD search. **e** Chemical structure of a molecule (CCDC reference code: JUHQEK) obtained from the CSD search. The amide carbonyl group is taken as CO-I and the ester carbonyl group is taken as CO-II here. **f** NBO orbital overlap between oxygen lone pair ($n_O$) of CO-I and $\pi^{\star}_{C=O}$ orbital of CO-II of JUHQEK. **g** NBO orbital overlap between oxygen lone pair ($n_O$) of CO-II and $\pi^{\star}_{C=O}$ orbital of CO-I of JUHQEK. [$d_1 = O^2 \cdots C^5$; $d_2 = O^6 \cdots C^1$; $\theta_1 = \angle O^2 \cdots C^5 = O^6$; $\theta_2 = \angle O^6 \cdots C^1 = O^2$ (Supplementary Fig. 3)]. Curved dotted arrows indicate n→π* interactions

which indicate their pyramidalization towards the donor oxygen atoms. The plots of Θ with O···C distances and the strength of the reciprocal interactions in compounds obtained from the CSD search are shown in Supplementary Fig. 5. Although the correlation between pyramidality of second carbonyl (CO-II) carbon ($\Theta_2$) and $d_1$ looks better than the correlation between pyramidality of first carbonyl (CO-I) carbon ($\Theta_1$) and $d_2$, the CO-I and CO-II are chosen completely randomly in these molecules. As the pyramidalization also depends on other factors like $\theta$ and the elasticity of the carbonyl group, a strong correlation between pyramidalization and the O···C distance and strength of n→π* interactions may not be observed in these molecules having different types of carbonyl groups as well as different $\theta$ values.

To get some insights into the structures of the small molecules having reciprocal C=O···C=O interactions, we manually analyzed small molecules from the CSD having 1,5-type reciprocal interactions with both $d_1$ and $d_2 \leq 3.00$ Å. A total of 249 molecules fulfill the above criteria [1, 5-interaction; both $d_1$ and $d_2 \leq 3.00$ Å]. As can be anticipated, the nature of the two atoms/ groups between the interacting carbonyl groups plays a key role in keeping the two carbonyl groups non coplanar and provides them the conformation required for reciprocal interactions (Supplementary Table 8). Interestingly, majority of these molecules (117, ~47%) have one heteroatom and one chiral carbon between the two interacting carbonyl pairs, a feature that resembles peptides and proteins.

**Table 2 X-ray crystallographic structural and NBO data of CSD molecules**

| CCDC Ref. code | $d_1$ | $d_2$ | $\theta_1$ | $\theta_2$ | $\Delta_1$ | $\Delta_2$ | $\Theta_1$ | $\Theta_2$ | $n \rightarrow \pi^*$ (kcal mol$^{-1}$) | | Total $n \rightarrow \pi^*$ (kcal mol$^{-1}$) |
|---|---|---|---|---|---|---|---|---|---|---|---|
| | (Å) | (Å) | (°) | (°) | (Å) | (Å) | (°) | (°) | $E^1_{(n \rightarrow \pi^*)}$ | $E^2_{(n \rightarrow \pi^*)}$ | $E^t_{(n \rightarrow \pi^*)}$ |
| PHTHAC05 | 2.997 | 2.996 | 72.0 | 72.1 | 0.031 | 0.032 | 3.76 | 3.86 | 0.12 | 0.13 | 0.25 |
| PODHUM | 3.068 | 3.062 | 72.3 | 72.6 | −0.007 | −0.025 | −0.88 | −2.96 | 0.07 | 0.04 | 0.11 |
| GECYEU | 2.992 | 2.972 | 74.7 | 75.5 | 0.032 | 0.009 | 3.92 | 1.10 | 0.31 | 0.16 | 0.47 |
| LEBRER | 2.861 | 2.856 | 77.3 | 77.6 | 0.006 | 0.021 | 0.67 | 2.45 | 0.77 | 0.60 | 1.37 |
| KOXBIK | 2.959 | 3.073 | 79.0 | 73.8 | −0.012 | 0.015 | −1.43 | 1.80 | 0.32 | 0.12 | 0.44 |
| CAJVIU | 3.006 | 2.987 | 77.6 | 78.5 | −0.004 | 0.023 | −0.43 | 2.68 | 0.41 | 0.12 | 0.53 |
| AZULUD | 2.979 | 3.009 | 80.0 | 78.6 | −0.013 | 0.011 | −1.52 | 1.33 | 0.36 | 0.20 | 0.56 |
| ZUKVUY | 2.887 | 2.829 | 81.5 | 84.2 | 0.006 | 0.027 | 0.74 | 3.24 | 0.86 | 0.34 | 1.20 |
| GAPDIK | 3.164 | 3.107 | 82.6 | 85.7 | 0.011 | 0.014 | 1.26 | 1.73 | 0.06 | 0.23 | 0.29 |
| LAGTIX | 3.137 | 3.119 | 82.7 | 83.0 | 0.008 | 0.001 | 0.94 | 0.06 | 0.18 | 0.06 | 0.24 |
| SUDAXAS01 | 3.171 | 3.101 | 82.7 | 85.9 | −0.013 | 0.008 | −1.42 | 0.88 | 0.21 | 0.08 | 0.29 |
| JUHQEK | 2.836 | 2.839 | 83.1 | 82.6 | 0.022 | 0.006 | 2.73 | 0.70 | 0.92 | 0.91 | 1.83 |
| ACBZO01 | 3.042 | 3.016 | 83.4 | 84.6 | −0.008 | 0.001 | −0.90 | 0.11 | 0.21 | 0.27 | 0.48 |
| WOCHIF | 2.885 | 2.832 | 83.1 | 85.3 | −0.009 | −0.011 | −1.13 | −1.28 | 0.55 | 0.85 | 1.40 |
| MODYIO | 3.136 | 3.102 | 83.8 | 85.5 | 0.003 | 0.005 | 0.35 | 0.58 | 0.15 | 0.17 | 0.32 |
| YEXQOH | 3.191 | 3.118 | 83.8 | 87.8 | −0.003 | 0.004 | −0.44 | 0.49 | 0.06 | 0.15 | 0.21 |
| CIQNEW | 3.044 | 3.086 | 84.1 | 82.2 | 0.001 | 0.001 | 0.15 | 0.14 | 0.23 | 0.20 | 0.43 |
| LUCHEY | 2.959 | 2.987 | 84.4 | 82.9 | −0.012 | 0.025 | −1.49 | 2.92 | 0.57 | 0.24 | 0.81 |
| BECLAW | 2.816 | 2.894 | 85.3 | 81.3 | −0.054 | 0.005 | −6.53 | 0.62 | 1.14 | 0.56 | 1.70 |
| DESPAT | 3.084 | 2.883 | 86.6 | 96.3 | 0.008 | 0.010 | 0.89 | 1.11 | 0.19 | 0.98 | 1.17 |
| GIRQAA | 3.032 | 2.998 | 90.7 | 92.1 | −0.005 | 0.013 | −0.64 | 1.55 | 0.32 | 0.36 | 0.68 |
| PUFBEZ | 3.132 | 3.107 | 93.9 | 94.9 | −0.036 | 0.002 | −4.46 | 0.26 | 0.08 | 0.20 | 0.28 |
| JOSGIH | 2.994 | 2.993 | 95.0 | 95.1 | 0.027 | 0.027 | 3.39 | 3.31 | 0.26 | 0.30 | 0.56 |
| IKAXII | 3.005 | 3.026 | 95.2 | 94.5 | 0.005 | −0.004 | 0.63 | −0.47 | 0.30 | 0.18 | 0.48 |
| OPAKIA | 2.985 | 3.008 | 95.3 | 93.4 | 0.005 | −0.020 | 0.58 | −2.39 | 0.23 | 0.34 | 0.57 |
| OMINII | 2.987 | 3.088 | 95.4 | 90.7 | 0.001 | 0.015 | 0.06 | 1.79 | 0.33 | 0.20 | 0.53 |
| XACLUK | 2.956 | 2.909 | 96.7 | 99.2 | −0.009 | 0.00 | −1.17 | 0.03 | 0.30 | 0.38 | 0.68 |
| EZELOK01 | 3.058 | 3.158 | 100.1 | 95.2 | −0.006 | 0.000 | −0.72 | −0.02 | 0.18 | 0.10 | 0.28 |
| WIHKAB | 2.866 | 3.054 | 103.2 | 93.4 | 0.012 | 0.004 | 1.31 | 0.58 | 0.57 | 0.04 | 0.61 |
| LOVNIT | 3.003 | 3.068 | 105.7 | 101.8 | 0.011 | −0.001 | 1.27 | −0.11 | 0.16 | 0.11 | 0.27 |

The calculations were carried out at B3LYP/6-311 + G(2d,p) level of theory. $d_1 = O^2 \cdots C^5$; $d_2 = O^6 \cdots C^1$; $\theta_1 = \angle O^2 \cdots C^5 = O^6$; $\theta_2 = \angle O^6 \cdots C^1 = O^2$ (Supplementary Fig. 3), $E^1_{(n \rightarrow \pi^*)}$, $E^2_{(n \rightarrow \pi^*)}$ and $E^t_{(n \rightarrow \pi^*)}$ have same meaning as described before in Table 1. [CO-I and CO-II are randomly chosen in these molecules]

**Reciprocal carbonyl-carbonyl interactions in proteins**. The presence of reciprocal C=O···C=O interactions in the X-ray crystal geometries of small organic molecules inspired us to look for their presence in protein crystal structures. To probe the presence of reciprocal C=O···C=O interactions in proteins, we analyzed a total of 2269 protein crystal structures with resolution ≤ 1.6 Å from the PDB with redundancy (pairwise sequence identity) less than 10%, out of which 2184 showed the presence of reciprocal interactions in them. The PDB protein structures ranked by the number of reciprocal C=O···C=O interactions present in them are included in Supplementary Data 2. For the PDB search, the distance between the carbonyl oxygen of $i$th amino acid residue and the carbonyl carbon of $(i + 1)$th amino acid residue is defined as $d_1$. The distance between the carbonyl oxygen of $(i + 1)$th residue and carbonyl carbon of $i$th residue is defined as $d_2$. The corresponding $\angle O \cdots C = O$ angles are defined as $\theta_1$ and $\theta_2$, respectively (Supplementary Fig. 6). During the search, both $d_1$ and $d_2$ were kept ≤ 3.20 Å but no restriction was imposed on $\theta_1$ and $\theta_2$. The plot of $d_1$ and $d_2$ values obtained from the search show that most of them fall in 2.90–3.20 Å range (Fig. 4a). The angles $\theta_1$ and $\theta_2$ (~85 ± 15°) deviates significantly from the Bürgi-Dunitz trajectory (Fig. 4b). These observations are consistent with the trend that was observed for small molecules discussed above. Analyses of $d_1$ and $d_2$ for all amino acid residues in all proteins (2184) studied here show that shorter distances $d_1$ and $d_2 \leq 3.2$ Å fall within the tail of the full distribution (Supplementary Fig. 7).

In a previous study[22], Bartlett et al reported one-sided $n \rightarrow \pi^*$ interactions with $d \leq 3.20$ Å and $99° \leq \theta \leq 119°$. As we have applied the same distance ($d \leq 3.20$ Å) and resolution (<1.6 Å) criteria, the reciprocal interactions observed here for angles $99° \leq \theta_1$, $\theta_2 \leq 119°$ would be observed as one-sided $n \rightarrow \pi^*$ interactions by using the criteria of Bartlett et al. As can be seen from Fig. 4b, the distribution of $\theta_1$ and $\theta_2$ in the range of 99°–119° (regions II, III, and IV) is a very small percentage (6.5%) of the total number of reciprocal C=O···C=O interactions that are being reported here. This indicates that reciprocal C=O···C=O interactions are novel and distinct from one-sided $n \rightarrow \pi^*$ interactions reported previously.

NBO analysis of 30 amino acid pairs (Supplementary Fig. 8) with short O···C distances (both $d_1$ and $d_2 \leq 3.20$ Å) that covers the complete range of observed $\angle O \cdots C = O$ angle ($\theta$) (70–110°) clearly showed the presence of reciprocal $n \rightarrow \pi^*$ interactions (Table 3, Figs. 4c, d). Similar to CSD molecules, in proteins also we observed substantial C=O···C=O $\pi \rightarrow \pi^*$ interactions between the amino acid pairs having $\theta_1$ and $\theta_2$ values > 90° (Supplementary Table 9). $\pi \rightarrow \pi^*$ NBO orbital overlap between the two carbonyl groups in an amino acid pair is shown in Figs. 4e, f [Leu-Pro (141–142); [PDB: 2x5o]. For molecules having relatively stronger $n \rightarrow \pi^*$ interactions (both $d_1$ and $d_2 < 2.90$ Å), $\pi \rightarrow \pi^*$ interactions were observed for $\theta_1$ and $\theta_2$ values <90° also (Table 3). This indicates that the overall contribution of reciprocal interactions (two $n \rightarrow \pi^*$ and two $\pi \rightarrow \pi^*$ interactions) could be substantial to protein stabilization. Based on the NBO calculations at B3LYP/6-311 + G(2d,p) level, we observed that reciprocal C=O···C=O interactions contribute 0.27–4.41 kcal mol$^{-1}$ (with an average value of 1.34 kcal mol$^{-1}$) to the stabilization of proteins per amino acid pair (see the last column in Supplementary Table 9).

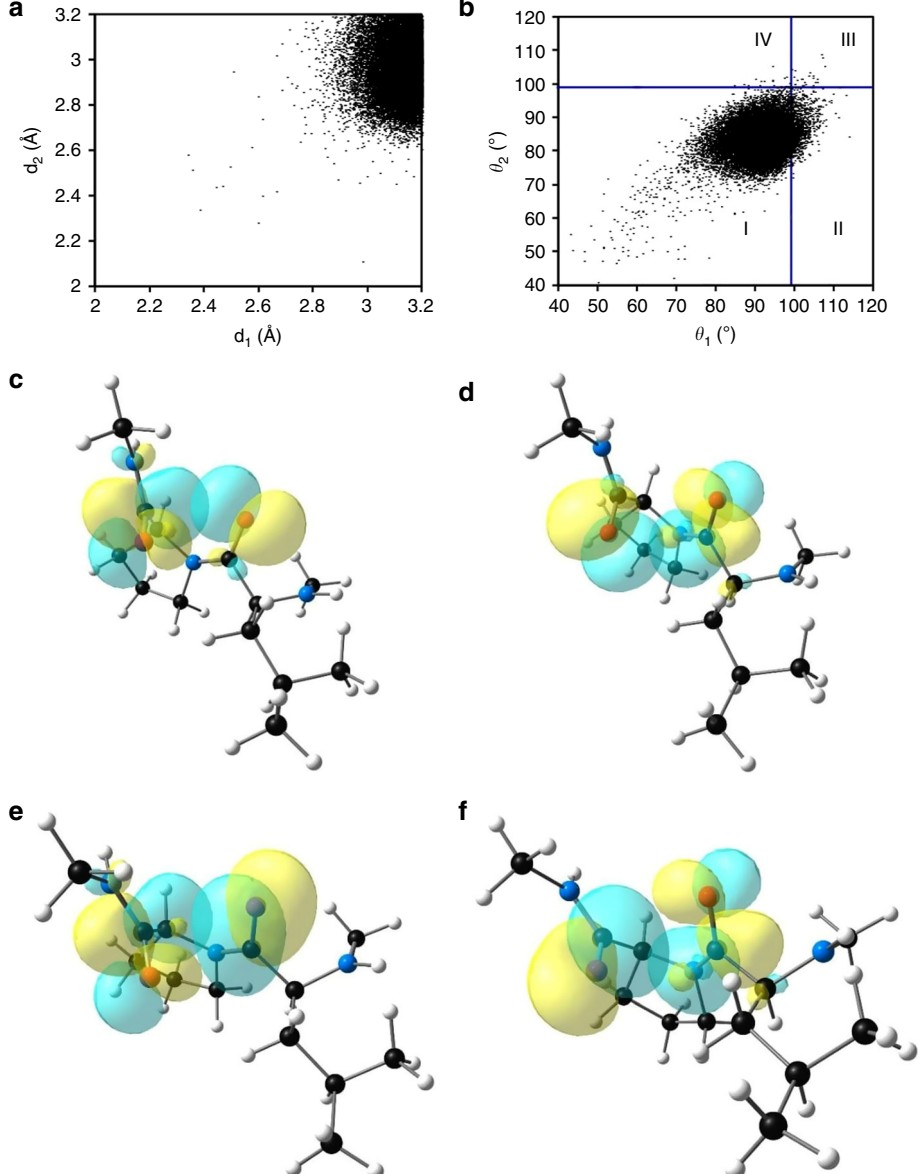

**Fig. 4** X-ray crystallographic data and NBO overlap diagrams for amino acid pairs. **a** Plot showing the distribution of O···C distances ($d_1$ and $d_2$) in amino acid pairs in proteins having reciprocal C=O···C=O interactions. **b** Plot showing the distribution of ∠O···C = O angles $\theta_1$ and $\theta_2$ in amino acid pairs in proteins having reciprocal C=O···C=O interactions. The *vertical* and *horizontal blue lines* are drawn at $\theta_1 = 99°$ and $\theta_2 = 99°$, respectively. **c** NBO orbital overlap between oxygen lone pair ($n_O$) of CO-I and $\pi^*_{C=O}$ orbital of CO-II of Leu-Pro (141-142) [PDB: 2x5o]. **d** NBO orbital overlap between oxygen lone pair ($n_O$) of CO-II and $\pi^*_{C=O}$ orbital of CO-I of Leu-Pro (141-142) [PDB: 2 × 5o]. **e** NBO orbital overlap between the $\pi$ orbital of C = O bond of CO-I and $\pi^*_{C=O}$ orbital of CO-II of Leu–Pro (141-142) [PDB: 2x5o]. **f** NBO orbital overlap between the $\pi$ orbital of C = O bond of CO-II and $\pi^*_{C=O}$ orbital of CO-I of Leu-Pro (141-142) [PDB: 2x5o]

The plot of torsion angles ($\varphi$, $\psi$) (Supplementary Fig. 6) of the residue between the two interacting carbonyl groups along with other residues in the proteins show that the reciprocal interactions are mainly concentrated in the polyproline II (PPII), β-turn and right-twisted β-strand regions (Fig. 5a). Unlike the one-sided $n \rightarrow \pi^*$ interactions reported previously[22, 23] that are abundant in proteins, the abundance of these newly discovered reciprocal C=O···C=O interactions is low (~7.2%). Secondary structure analyses using Stride[38] show that reciprocal C=O···C=O interactions have considerable abundance in random coils (~20%) and turn regions (10%) of proteins but negligible presence in α-helices (0.35%) (Table 4). This is in contrast to the one-sided $n \rightarrow \pi^*$ interactions that are most abundant in α-helices[22, 23]. As PPII helix is not included as an independent

secondary structure in most secondary structure predication programs, many PPII helices remain unassigned even though they are present in the experimentally solved structures. We observed that the coil regions having reciprocal C=O···C=O interactions are dominated by PPII structures [($\varphi$, $\psi$) : (−75°, 145°)]. We have confirmed this by plotting the $\varphi$, $\psi$ angles of residues in the random coil regions having reciprocal interactions (Fig. 5b). This is not surprising given that PPII conformations are known to dominate coil regions of folded proteins[39].

We also manually analyzed 789 reciprocal C=O···C=O interactions in 10 proteins having the highest numbers of reciprocal C=O···C=O interactions (Supplementary Table 10). In agreement with Stride prediction, manual inspection revealed that reciprocal C=O···C=O interactions are mostly present in

**Table 3 X-ray crystallographic structural and NBO data for amino acid pairs from the PDB**

| Amino acid pair | PDB code | Residues | $d_1$ | $d_1$ | $\theta_1$ | $\theta_2$ | n→π* (kcal mol⁻¹) | | Total n→π* (kcal mol⁻¹) |
|---|---|---|---|---|---|---|---|---|---|
| | | | (Å) | (Å) | (°) | (°) | $E^1_{(n\to\pi^*)}$ | $E^2_{(n\to\pi^*)}$ | $E^t_{(n\to\pi^*)}$ |
| Ile–Pro | 2opc | 135–136 | 2.675 | 2.768 | 75.6 | 71.8 | 1.75 | 0.74 | 2.49 |
| Lys–Pro | 1k3i | 50–51 | 2.975 | 2.938 | 78.9 | 80.5 | 0.44 | 0.38 | 0.82 |
| Cys–Pro | 1gcy | 251–252 | 2.815 | 2.834 | 80.6 | 79.8 | 1.24 | 0.60 | 1.84 |
| Leu–Pro | 1g5a | 379–380 | 2.956 | 2.978 | 80.7 | 79.6 | 0.61 | 0.27 | 0.88 |
| Ile–Pro | 1o7i | 107–108 | 2.986 | 3.082 | 81.6 | 77.8 | 0.44 | 0.19 | 0.63 |
| Val–Pro | 1jnd | 294–295 | 3.119 | 3.135 | 81.6 | 81.1 | 0.17 | 0.10 | 0.27 |
| Ala–Pro | 2xu9 | 264–265 | 2.568 | 2.852 | 83.8 | 72.0 | 3.63 | 0.56 | 4.19 |
| Thr–Pro | 1fj2 | 3–4 | 3.035 | 3.015 | 84.2 | 85.4 | 0.26 | 0.34 | 0.60 |
| Ile–Pro | 1e2w | 208–209 | 3.114 | 3.143 | 84.4 | 83.2 | 0.24 | 0.07 | 0.31 |
| Leu–Pro | 1a2p | 20–21 | 2.904 | 3.007 | 85.3 | 80.5 | 0.72 | 0.33 | 1.05 |
| Pro–Pro | 3cx2 | 186–187 | 2.890 | 2.838 | 86.3 | 88.7 | 0.98 | 0.39 | 1.37 |
| Glu–Pro | 1eu1 | 623–624 | 3.144 | 3.194 | 88.0 | 85.8 | 0.22 | 0.03 | 0.25 |
| Leu–Pro | 2x5o | 141–142 | 2.712 | 2.771 | 88.9 | 86.5 | 1.58 | 1.30 | 2.88 |
| Ala–Pro | 1g12 | 103–104 | 3.108 | 3.152 | 89.3 | 87.3 | 0.32 | 0.10 | 0.42 |
| Pro–Ser | 4psc | 32–33 | 2.998 | 2.946 | 90.4 | 92.4 | 0.46 | 0.33 | 0.79 |
| Thr–Glu | 4pdy | 108–109 | 2.918 | 2.936 | 91.5 | 90.4 | 0.72 | 0.25 | 0.97 |
| His–Ser | 1b6a | 331–332 | 3.128 | 3.108 | 92.3 | 93.2 | 0.13 | 0.14 | 0.27 |
| Ala–Asp | 2bi8 | 95–96 | 2.985 | 2.973 | 92.8 | 93.3 | 0.52 | 0.13 | 0.65 |
| Phe–Pro | 1n08 | 76–77 | 2.943 | 2.941 | 93.2 | 93.3 | 0.52 | 0.10 | 0.62 |
| Leu–Pro | 1eb6 | 110–111 | 3.176 | 3.175 | 93.4 | 93.2 | 0.14 | 0.10 | 0.24 |
| Leu–Tyr | 3u26 | 101–102 | 2.826 | 2.893 | 93.6 | 90.2 | 0.96 | 0.34 | 1.30 |
| Ser–Asp | 3ry4 | 79–80 | 2.952 | 2.981 | 94.2 | 92.9 | 0.56 | 0.12 | 0.68 |
| Ala–Phe | 4y1w | 139–140 | 2.947 | 2.973 | 95.3 | 93.9 | 0.54 | 0.26 | 0.80 |
| Gln–Lys | 3wcq | 9–10 | 2.815 | 2.816 | 95.9 | 96.9 | 1.03 | 0.16 | 1.19 |
| Thr–Thr | 3uxf | 349–350 | 2.987 | 2.975 | 96.4 | 97.3 | 0.28 | 0.13 | 0.42 |
| Ala–Arg | 1ejd | 119–120 | 3.175 | 3.171 | 99.3 | 99.0 | 0.13 | NP | 0.13 |
| Phe–Gly | 1odv | 28–29 | 3.129 | 3.169 | 102.4 | 100.3 | 0.13 | 0.02 | 0.15 |
| Asp–Pro | 2vzp | 2–3 | 3.026 | 3.026 | 103.6 | 103.3 | 0.07 | 0.19 | 0.26 |
| Ala–Leu | 1ikp | 388–389 | 3.141 | 3.183 | 104.5 | 102.4 | 0.12 | NP | 0.12 |
| Ala–Ala | 3s5m | 402–403 | 2.485 | 2.999 | 113.6 | 84.2 | 0.89 | 0.01 | 0.90 |

The calculations were carried out at B3LYP/6-311 + G(2d,p) level of theory. For the definitions of $d_1$, $d_2$, $\theta_1$ and $\theta_2$ see Supplementary Fig. 6. $E^1_{(n\to\pi^*)}$, $E^2_{(n\to\pi^*)}$ and $E^t_{(n\to\pi^*)}$ have same meaning as described before in Table 1. [CO-I and CO-II are randomly chosen in these molecules]
NP not present

coil/PPII and turn regions of these proteins. α-helices that have reciprocal C=O⋯C=O interactions are distorted, while the β-sheets having reciprocal n→π* interactions are twisted (Fig. 6). We also observed reciprocal C=O⋯C=O interactions between amino acid pairs at the interfaces of different secondary structure types (Fig. 6e and Supplementary Table 11).

The other secondary structure that has significant abundance of reciprocal C=O⋯C=O interactions is β-turn. In a β-turn, the peptide groups (NH and C = O) of the central two amino acids do not participate in any inter-residue hydrogen bonding. Therefore, we assume that these residues may participate in local reciprocal C=O⋯C=O interactions either between themselves or with their other neighbors, which should compensate for the lack hydrogen bonding interactions in them. A careful examination of the orientations of the carbonyl groups in various common β-turns indicated that reciprocal C=O⋯C=O interaction may be feasible between the first and the second residues of type I′ and II β-turns due to the favorable orientations of the two carbonyl groups but likely to be unfavoured in type I and II′ β-turns. In fact, analysis of the 10 protein crystal structures discussed above show that most of the reciprocal C=O⋯C=O interaction pairs found in β-turns were type II, followed by type IV (Supplementary Table 12). In 41 cases, the reciprocal n→π* interactions were present between first and the second amino acid residues while in other 43 cases they were between the third and the fourth amino acid residues of β-turns in these 10 proteins. However, in no case the second and the third residues of the β-turn were involved in reciprocal C=O⋯C=O interactions between them (Supplementary Fig. 9).

Analysis of distribution of reciprocal C=O⋯C=O interactions among various amino acids suggests that proline is involved in the largest number of reciprocal C=O⋯C=O interactions in various proteins followed by glutamic acid and serine (Fig. 5c). This trend is different from what was previously observed for one-sided n→π* interactions in α-helices and β-sheets[22] (Pro > Gly > Ala). Analysis of distribution of reciprocal C=O⋯C=O interactions among the amino acid pairs in various proteins reveals that Pro–Pro is the most abundant pair (Fig. 5d). The 10 most prominent amino acid pairs that participate in reciprocal C=O⋯C=O interactions, all contain a proline residue (Fig. 5d). These results may be expected given the abundance of reciprocal interactions in PPII regions.

**Possible role of reciprocal carbonyl-carbonyl interactions in protein folding.** PPII helices and turns are the major secondary structures where reciprocal C=O⋯C=O interactions are observed. PPII is the major well-defined backbone structure present in denatured, unfolded, and natively unfolded proteins[40] and random coil regions of folded proteins[39]. As PPII lacks stable non-local amino acid interactions such as hydrogen bonding, we propose that local reciprocal interactions could possibly contribute to their stability. Levinthal proposed that protein folding is speeded and guided by the rapid formation of local interactions in the unfolded state, which then determine the further folding of the peptide[41]. The fact that local reciprocal interactions contribute to the stabilization of the PPII conformation that are abundant in unfolded proteins, reciprocal

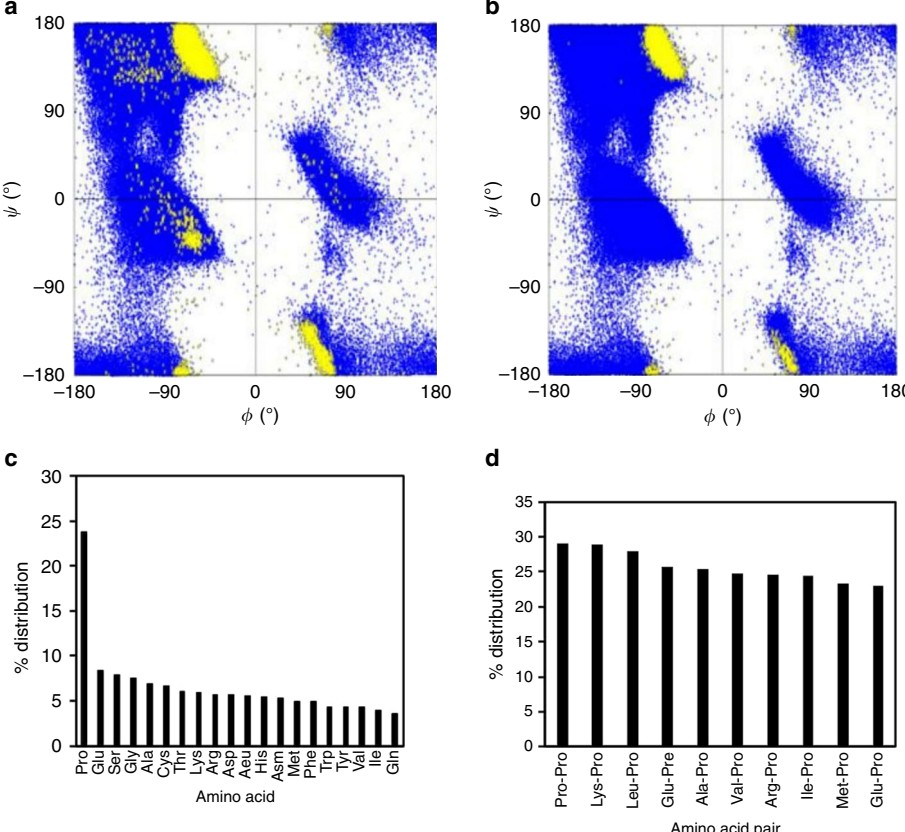

**Fig. 5** Ramachandran plots and analyses of reciprocal interactions in proteins. **a** Ramachandran plot generated by plotting torsion angles ($\varphi$, $\psi$) of all residues in 2184 protein structures (*blue*) and torsion angles ($\varphi$, $\psi$) of the residue between the two interacting carbonyl groups involved in reciprocal C=O···C=O interactions (*yellow*). **b** Ramachandran plot generated by plotting torsion angles ($\varphi$, $\psi$) of all residues in 2184 protein structures (*blue*) and torsion angles ($\varphi$, $\psi$) of the residue between the two interacting carbonyl groups involved in reciprocal C=O···C=O interactions present only in the coil regions (*yellow*). **c** Plot showing percentage distribution of amino acids involved in reciprocal C=O···C=O interactions. **d** Plot showing percentage distribution of amino acid pairs involved in reciprocal C=O···C=O interactions

**Table 4 Distribution of reciprocal carbonyl-carbonyl interactions in various secondary structures**

| Secondary structure type | Total number of amino acids | Amino acids involved in reciprocal C=O···C=O interactions | % |
|---|---|---|---|
| Coil | 93743 | 18422 | 19.65 |
| Turn | 124938 | 12577 | 10.07 |
| β-sheet | 131564 | 6195 | 4.71 |
| 3₁₀-helix | 22712 | 408 | 1.80 |
| α-helix | 156248 | 541 | 0.35 |
| Overall | 529205 | 38143 | 7.21 |

interactions could play a role in protein folding. Also, turn regions that are stabilized by reciprocal interactions are known to act as nucleation sites for protein folding. Therefore, an open question is how important such reciprocal interactions might be for protein folding.

**Nature of reciprocal carbonyl-carbonyl interactions.** The nature of C=O···C=O interactions has been debated in the literature. While some consider them $n{\rightarrow}\pi^{*}$ orbital interactions[9, 11], others believe them to be dipolar in nature[7, 8, 10]. We have so far

discussed reciprocal C=O···C=O interactions as $n{\rightarrow}\pi^{*}$ and $\pi{\rightarrow}\pi^{*}$ orbital interactions because of the following reasons. Firstly, the plots of the $n{\rightarrow}\pi^{*}$ and sum of $n{\rightarrow}\pi^{*}$ and $\pi{\rightarrow}\pi^{*}$ orbital interaction energies against the O···C distances (d) show a strong correlation (Figs. 7a, b). In Fig. 7a, we have plotted the distances ($d_1$ and $d_2$ values) against the stabilization energies due to $n{\rightarrow}\pi^{*}$ interactions [NBO second order perturbation energies $E^1{}_{(n{\rightarrow}\pi^*)}$ and $E^2{}_{(n{\rightarrow}\pi^*)}$] reported in Tables 1–3. The plot suggests that the stabilization energies $E_{(n{\rightarrow}\pi^*)}$ for $n{\rightarrow}\pi^{*}$ interactions decreases with an increase in the O···C (d) in synthetic molecules **1**–**8**, molecules taken from CSD and interacting amino acid pairs obtained from PDB (Tables 1–3). The overall orbital interaction energies (sum of $n{\rightarrow}\pi^{*}$ [$E_{(n{\rightarrow}\pi^*)}$] and $\pi{\rightarrow}\pi^{*}$ [$E_{(\pi{\rightarrow}\pi^*)}$] interaction energies reported in Tables 1–3) plotted in Fig. 7b also show a similar correlation with O···C (d) distances. These correlations indicate that orbital interaction is the major mechanism for the stabilization of these reciprocal C=O···C=O short contacts. Secondly, we carried out NBO deletion analysis on all the molecules reported in Tables 1–3 (Supplementary Table 13) and observed that deletion of $n{\rightarrow}\pi^{*}$ interactions increases charge on donor oxygen lone pair ($n_O$) and depletes it on acceptor carbonyl $\pi^{*}{}_{C=O}$ orbital, which correlate well with the strength of O···C distances (Supplementary Fig. 10a, b). Similarly, deletion of $\pi{\rightarrow}\pi^{*}$ interactions increases charge on $\pi_{C=O}$ orbital of donor carbonyl and depletes it on $\pi^{*}{}_{C=O}$ orbital of the acceptor carbonyl (Supplementary Table 14), which also can be correlated to the strength of C=O···C=O short contacts (Supplementary Fig. 11a, b). The overall accumulation of

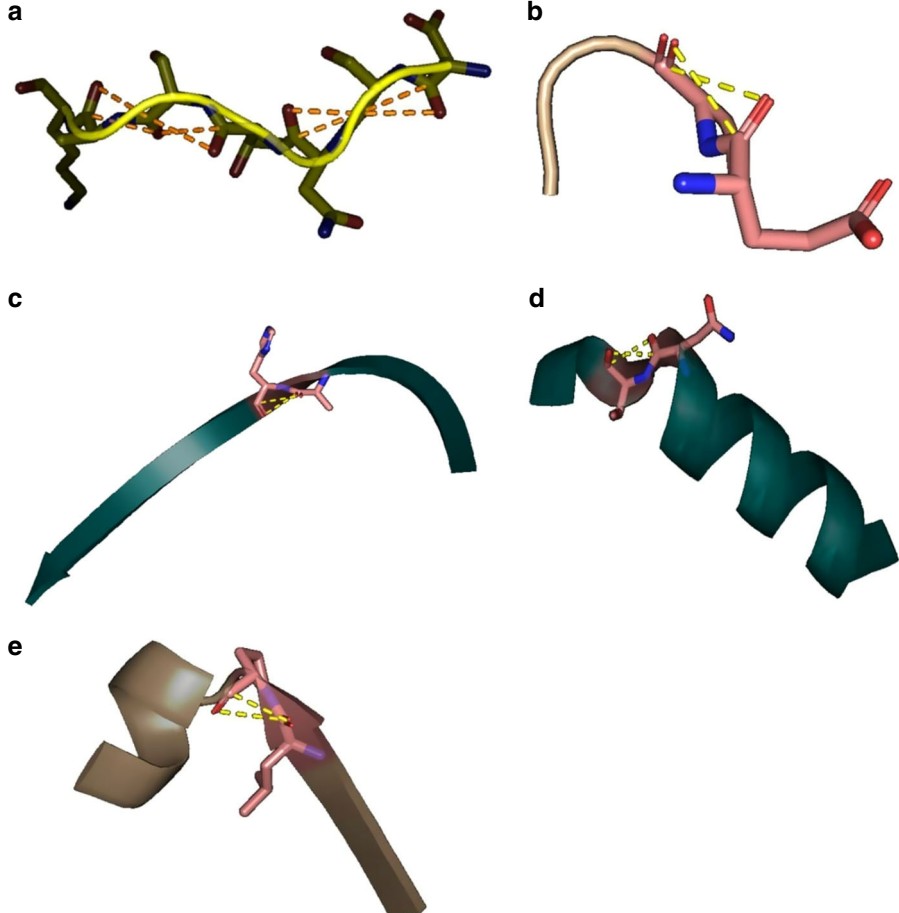

**Fig. 6** Reciprocal carbonyl-carbonyl interactions in various secondary structures. **a** PPII-helix; **b** β-turn; **c** Right-twisted β-strand; **d** α-helix; **e** interface of α-helix and β-sheet. The Figures are generated by using PyMOL

charges on the acceptor carbonyl $\pi^{\star}_{C=O}$ orbitals due to donation from the oxygen lone pairs and $\pi_{C=O}$ orbital of donor carbonyl is shown in Figs. 7c–d, which correlate well with the strength of $C=O\cdots C=O$ short contacts. This also suggests that electron delocalization is a major contributor in reciprocal $C=O\cdots C=O$ interactions. Finally, $C=O\cdots C=O$ torsion angles of the carbonyl groups involved in reciprocal interactions indicate a net zero dipole-dipole interaction eliminating the possibility of these interactions being dipolar in nature. To emphasize this point, in Figs. 7e–f, we have plotted the values of $C=O\cdots C=O$ torsion angles of the 1432 molecules obtained from the CSD search. The torsion angle ($T$) between two dipoles could be used to under-stand the dipolar nature of interaction between them. As we know, antiparallel ($T \sim 180°$) dipoles attract and parallel dipoles ($T \sim 0°$) repel each other whereas two orthogonal dipoles ($T \sim 90°$) have net zero dipolar interaction. In case of reciprocal interaction, the $C=O\cdots C=O$ torsion angles show an orienta-tional preference [$C=O\cdots C=O$ torsion angle falls in 60° to 90° (or −60° to −90°) range] as a consequence of the simultaneous restrictions on $d_1$ and $d_2$ (≤3.2 Å). However, the values of the $C=O\cdots C=O$ torsion angles ($\sim 90°$) suggest that there would be almost net zero interaction between the dipoles, eliminating the possibility of strong dipolar interactions. Therefore, we conclude that orbital delocalization is the major driving force for the sta-bilization of reciprocal $C=O\cdots C=O$ interactions. An elaborate energy decomposition analysis may be required for the accurate deconvolution of various factors contributing to the stabilization of reciprocal $C=O\cdots C=O$ short contacts.

We conclude that reciprocal carbonyl-carbonyl interactions exist both in small organic molecules and proteins. However, due to geometrical constraints associated with such interactions, the approach of the donor oxygen atoms to the acceptor carbon atoms deviates significantly from the Bürgi-Dunitz trajectory, and therefore, electron delocalization between the oxygen lone pair ($n_O$) and $\pi^{\star}_{C=O}$ orbital is weak. This weak donation from the first carbonyl group to the second is compensated by a back donation from the second carbonyl group to the first. In many cases, reciprocal $\pi\rightarrow\pi^{\star}$ interactions were also observed along with reciprocal $n\rightarrow\pi^{\star}$ interactions and their overall contributions to the stabilization of molecules having reciprocal $C=O\cdots C=O$ short contacts could be significant. In proteins, $C=O\cdots C=O$ $n\rightarrow\pi^{\star}$ interactions are present in all types of secondary structures. While one-sided $n\rightarrow\pi^{\star}$ interactions are prevalent in α-helices[22, 23], reciprocal interactions are abundant in PPII helices and turn regions. Prevalence of reciprocal $C=O\cdots C=O$ interactions in PPII helices and turn regions of proteins suggests a possible role for these interactions in protein folding. Further, the presence of reciprocal $C=O\cdots C=O$ interactions in distorted α-helices and twisted β-sheets suggests that these interactions could stabilize secondary structures that deviate from their regular geometries. The reciprocal $C=O\cdots C=O$ interactions present at the interface of two different types of secondary structures could also help in stabilizing the strained amino acid residues that are present at these interfaces. In future, it would be interesting to investigate the ability of amino acid pairs having high propensity to get involved in reciprocal $C=O\cdots C=O$ interactions to stabilize PPII

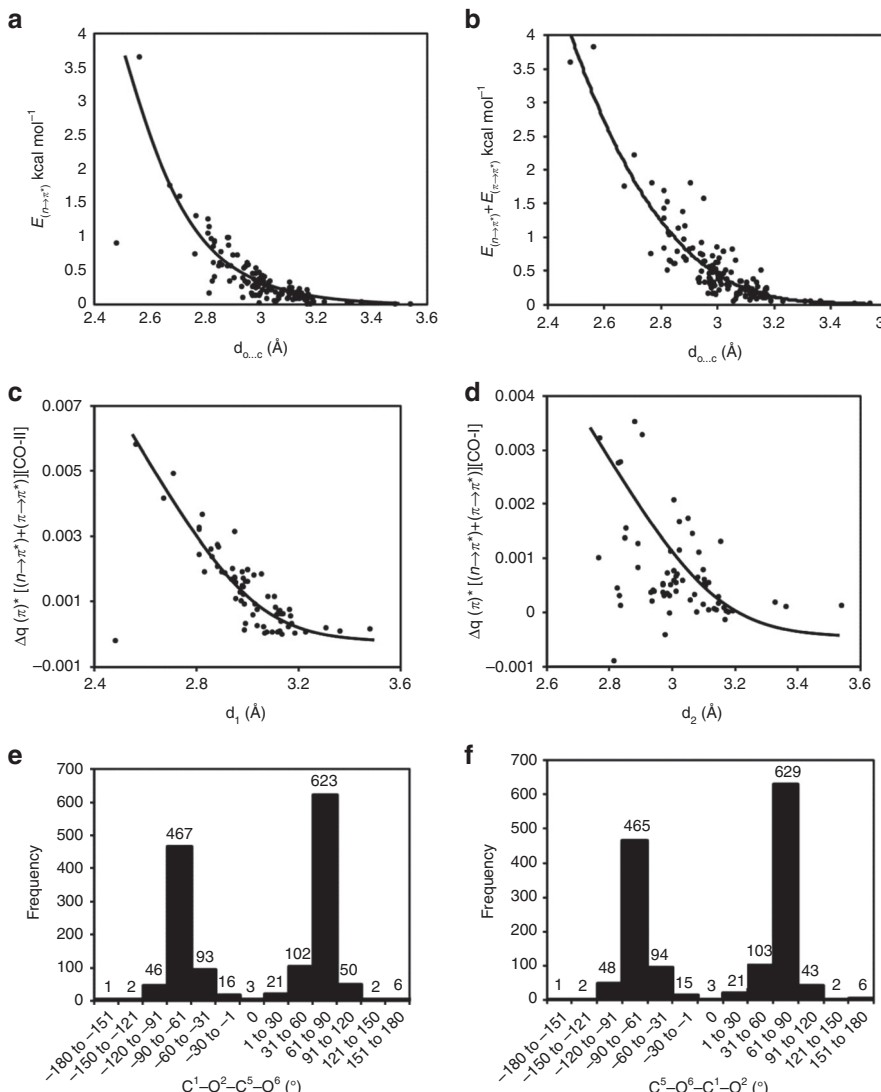

**Fig. 7** Delocalization energies, charge redistribution and torsion angles. **a** Plot of $n \to \pi^*$ interaction energies between the interacting carbonyl pairs against crystallographic O···C distances ($d_1$ and $d_2$) in molecules shown in Tables 1–3. When the x-axis is $d_1$, $E^1_{(n \to \pi^*)}$ is plotted in the y-axis and when the x-axis is $d_2$, $E^2_{(n \to \pi^*)}$ is plotted in the y-axis. The $d_1$, $d_2$, $E^1_{(n \to \pi^*)}$ and $E^2_{(n \to \pi^*)}$ values are taken from Tables 1–3. The $n \to \pi^*$ interaction energies were computed at B3LYP/6-311 + G(2d,p) level of theory. **b** Plot of overall orbital interaction energy (sum of $n \to \pi^*$ and $\pi \to \pi^*$ interaction energies) between the interacting carbonyl pairs against crystallographic O···C distances ($d_1$ and $d_2$) in molecules shown in Tables 1–3. When the x-axis is $d_1$, $E^1_{(n \to \pi^*)} + E^1_{(\pi \to \pi^*)}$ is plotted in the y-axis and when the x-axis is $d_2$, $E^2_{(n \to \pi^*)} + E^2_{(\pi \to \pi^*)}$ is plotted in the y-axis. $d_1$, $d_2$ $E^1_{(n \to \pi^*)}$ and $E^2_{(n \to \pi^*)}$, values are taken from Tables 1–3. $E^1_{(\pi \to \pi^*)}$ and $E^2_{(\pi \to \pi^*)}$ values are taken from Supplementary Table 3, Supplementary Table 7 and Supplementary Table 9. The orbital interaction energies were computed at B3LYP/6-311 + G(2d,p) level of theory. **c** Plot of accumulation of charges on the $\pi^*_{C=O}$ orbital of CO-II due to donation from lone pairs of oxygen and $\pi_{C=O}$ orbital of CO-I against $d_1$. **d** Plot of accumulation of charges on the $\pi^*_{C=O}$ orbital of CO-I due to donation from lone pairs of oxygen and $\pi_{C=O}$ orbital of CO-II against $d_2$. The solid curves in a-d are drawn for convenience. **e** Histogram plot showing the frequency of the $C^1=O^2 \cdots C^5=O^6$ dihedral angles (see Supplementary Fig. 3 for atom numbers) for 1432 molecules obtained from the CSD search. **f** Histogram plot showing the frequency of the $C^5=O^6 \cdots C^1=O^2$ dihedral angles (see Supplementary Fig. 3 for atom numbers) for 1432 molecules obtained from the CSD search

helices and β-turns. It would also be interesting to investigate if some non-peptidic fragments obtained from the CSD search having strong reciprocal C=O···C=O interactions could be used to stabilize PPII conformation or design peptide-turns. Finally, an energy decomposition analysis would provide better understanding of the forces that contributes to the stabilization of reciprocal C=O···C=O interactions.

## Methods

**Crystallization method**. Single crystals of compounds **1–8** were grown by slow evaporation. Various solvent combinations were used to crystallize the compounds either at room temperature or low temperature (4 °C). Details of the crystallization conditions are given in Supplementary Table 1.

**X-ray crystal structure determination method**. Single crystal structures of compound **1–8** were determined by measuring X-ray intensity data. Bruker D8Venture *APEX* 3[42] single crystal home source X-ray diffractometer equipped with CMOS PHOTON 100 detector and Monochromated microfocus sources Mo Kα radiation ($\lambda = 0.71073$ Å) were used for data collection in phi (ϕ) and omega (ω) scan strategy at room temperature (298 K). The data was processed using *SAINT*[43] and absorption correction was done using *SADABS*[44] implemented in *APEX* 3. For structure solution XSHELL program based on SHELX[45] was used. The non-hydrogen atoms were refined anisotropically and located in successive difference Fourier syntheses. The hydrogen atoms were fixed to neutron bond length using appropriate HFIX commands. ORTEP diagrams of compounds **1–8** (CCDC 1486577- 1486584) is provided in Supplementary Fig. 1. Compound **5** crystallized with a water molecule in the asymmetric unit. However, for clarity we have not shown the water molecule in its ORTEP diagram. Compound **7** has disorder at chlorine atom; the occupancy of disordered chlorine atom namely Cl1A

and Cl1B was refined using the PART command. Similar ADP restraint SIMU[46] and rigid bond restraint DELU[46] was applied to stabilize the anisotropic refinement. SADI[46] instruction was used to restrain the distance to equal. The anisotropic displacement parameter for disordered chlorine atom was fixed using EADP[46] constraint.

**CSD analysis.** Intramolecular C═O···C═O noncovalent interactions were searched and structural data were retrieved from Cambridge Structural Database[33] (CSD version 5.21 Nov. 2015) using Conquest[47] (version 1.18) program. The fragment chosen for the search is shown in Supplementary Fig. 3, where $X$ is indicative for any atom. Only unique matching fragments were taken and the fragment was chosen in such a way that there are at least two carbonyl groups irrespective of their nature. Distances $d_1$ ($O^2$–$C^5$) and $d_2$ ($O^6$–$C^1$) are restricted to $\leq 3.2$ Å. Angles [$O^2$–$C^5$–$O^6$ ($\theta_1$) and $O^6$–$C^1$–$O^2$ ($\theta_2$)] and dihedral angles ($C^1$–$O^2$–$C^5$–$O^6$ and $C^5$–$O^6$–$C^1$–$O^2$) were printed without any restriction. Only crystalline, non-ionic and non-polymeric organic molecules having no disorder and error with $R$ factor $\leq 5\%$ having at least three covalent bond separations between the carbonyl groups were considered in this search.

**PDB analysis.** A subset of 2269 protein was culled out from RCSB PDB[34] using a search criterion of resolution <1.6 Å with redundancy (pairwise sequence identity) less than 10%, downloaded on 19 January 2016. Out of 2269 proteins, 2184 showed the reciprocal $n\rightarrow\pi^*$ interaction. For proteins existing in polymeric form or for proteins containing amino acids in more than one conformation, Chain A and conformation A were chosen, except for 57 proteins where chain A is absent. Distance $d_1$ is defined as distance between the $i^{th}$ amide oxygen to the subsequent $(i+1)^{th}$ amide carbon, while $d_2$ is defined as distance between the $(i+1)^{th}$ amide oxygen to the $i^{th}$ amide carbon (Supplementary Fig. 6). We used $d_1 \leq 3.2$ Å and $d_2 \leq 3.2$ Å criteria for selecting amino acid pairs participating in reciprocal $n\rightarrow\pi^*$ interactions. Secondary structure assignment was done using the Stride code[39]. Ramachandran plots were generated for the proteins using Gnuplot (http://www.gnuplot.info/).

**Computational methods.** All the calculations were performed by using Gaussian09 suite of quantum chemistry programs[48]. The Hartree-Fock (HF)[49] and the hybrid Becke 3-Lee-Yang-Parr (B3LYP)[50, 51] exchange correlation functional with 6-311 + G (2d,p) basis set were used for the calculations. Natural bond orbital (NBO)[35] analyses were performed on the crystal geometries of the synthetic molecules and small organic molecules obtained from CSD search. For proteins, the coordinates of the interacting amino acid residue pair were extracted using PyMOL[52]. The α-carbons of the amino acid residues adjacent to $N$ and $C$ termini of the amino acid pair were also included, so as to mimic a dipeptide with $N$ and $C$ termini capped with N(CO)Me and (CO)NMe, respectively. Finally, hydrogen atoms were added to the structure using PyMOL (Supplementary Fig. 8). NBO analyses were carried out on crystal geometries at B3LYP/6-311 + G(2d,p) and HF/6-311 + G(2d,p) level of theory. The NBO second order perturbative energies $E_{(n\rightarrow\pi^*)}$ and $E_{(\pi\rightarrow\pi^*)}$ obtained from NBO calculations were taken as the stabilization energy due to $n\rightarrow\pi^*$ and $\pi\rightarrow\pi^*$ interactions. NBO deletion analysis was carried out on crystal geometries at HF/6-311 + G(2d,p) level of theory.

**Data availability.** The authors declare that the data supporting the findings of this study are available within the paper and its Supplementary Information files, and also are available from the corresponding author upon reasonable request. X-ray crystallographic data for structures reported in this study have been deposited at the Cambridge Crystallographic Data Centre (CCDC), under deposition number CCDC 1486577-1486584. These data can be obtained free of charge from the CCDC via www.ccdc.cam.ac.uk/.

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

## Acknowledgements

We thank E. Arunan, M. Goswami, B.B. Dhar, R.P. Vivekananth and R. Ayana for helpful discussions. This project was funded by Shiv Nadar University and Early Career Research Grant (ECR/2015/000337) from Science and Engineering Research Board (SERB), Department of Science and Technology (DST), Government of India.

## Author contributions

B.K.S. conceived the project. A.R. carried out the synthesis and characterization of the compounds. A.R. crystallized the compounds, K.K.J. collected the X-ray data and K.K.J. and A.R. solved the structures. B.K.S. and A.R. designed the CSD analyses. A.R. performed the CSD analyses. B.K.S. and P.S. designed the PDB analyses. P.S. performed the PDB analyses. B.K.S. designed the computational studies and A.R. performed them. B.K.S. wrote the manuscript. B.K.S., A.R., N.S., P.S., and K.K.J. discussed the results and edited the manuscript.

## Additional information

**Competing interests:** The authors declare no competing financial interests.

