## [Peer Review File · Nature Communications]

Reviewer #1 (Remarks to the Author):

Two-side $n \rightarrow n^*$ interactions in small molecules and proteins

This is an interesting paper on a potential new manifestation of the now largely accepted $n \rightarrow n^*$ interaction in small molecules and proteins. In short, the hypothesis is that in addition to the established one-sided interaction, two-sided interactions are possible between nearby carbonyl groups that are appropriately configured in space to allow what I would prefer to call "reciprocated $n \rightarrow n^*$ interactions". Impressively, the authors combine experimental studies in a small-molecule model, analysis of two databases (namely the CSD and the PDB), and theoretical calculations to make and support their hypothesis. I am left in no doubt that these two-sided $n \rightarrow n^*$ interactions do occur in small and protein molecules, and that they may well be important in contributing to stabilising the final conformations of small and protein molecules, though I am more sceptical about roles that they might play in protein folding. As a result, I would guardedly say that this could be published in Nat Commun. However, I would suggest that if the editors do decide to proceed that they ask for major revisions before accepting the paper. This is for several reasons.

First, whilst I am in no doubt that the detail, quality and depth of this analysis and paper far outstrips any foregoing work, there are precedents for describing this type of arrangement of carbonyl groups in the literature, regardless of the mechanism of interaction. See:

Allen et al., Acta Cryst B54: 320-29 (1998) (Figure 1).

MacCallum et al (Milner-White and co), JMB (1995), 248:374-384 (Figure 3 and text).

For the background and context of this new work, these foregoing papers must be cited and discussed in a revised manuscript.

Page 4. I am confused by the energies of interactions 1 and 2. These are in effect closed systems, and one might anticipate that the electrons become evenly distributed. So why the difference in energy between the "1st" and "2nd" $n \rightarrow n^*$ interaction? I'm not sure that I buy the explanation given. Moreover, the argument made does not tally with the data given in Table 2, where all possibilities are seen, interaction 1 > interaction 2, 2 > 1 and 1 \approx 2.

Figures 2 and 3. The distances d_1 and d_2 are described and shown for the systems that test positive for the two-sided $n \rightarrow n^*$ only. But what is the distribution of these distances in unconstrained systems. (I realise that this probably only has meaning in the context of $i, i+1$ residues in proteins.) This is important to see if the short distances are really outliers in the full distributions or not.

None of the data plotted in Figs. 4a-c are normalized, ie raw counts or frequencies are given rather than percentages, or, better, propensities. This is also important. It is best illustrated by example: Clearly, in panel a two-sided $n \rightarrow n^*$ are rare in alpha and beta structures, but we can't compare these numbers with coil and turn, because these states occur in different amounts across the whole PDB. Therefore, the numbers for each state need to be normalized somehow for the % of that state in the PDB as a whole. Numbers could then be compared. This will help the authors' argument I'm sure, as coil is rarer than alpha etc.

The same is true for the amino acids in panel b. Met, Cys and Trp are the amino acids that occur least frequently in proteins, so it is no surprise that they sit on the RHS of the plot. I suspect that normalising for amino acid occurrence in the PDB would flatten all amino acids other than Pro and Gly, which would add weight to the authors' argument.

While on this, panel c seems redundant as it only highlights the Pro and Gly-containing pairs really. Panel d is hard to fathom and explain, and should be removed in my view.

While on Pro, have the authors considered cis/trans forms and how these differ in propensity to form two-sided $n \rightarrow n^*$ interactions?

I mentioned my preference for "reciprocated $n \rightarrow n^$ interactions" as I find "two-sided $n \rightarrow n^*$ interactions" confusing. The authors may wish to consider this alternative name.

Finally, I am not taken by the arguments that these could influence folding. Even at $\frac{1}{4}$ kcal/mol for each $n \rightarrow n^*$ interaction these would perturb equilibria between states only slightly. This is before solvent is considered. Surely, water competes well for solvation of carbonyl groups in a flexible unfolded state? I would recommend that this section is removed or trimmed at least. In other words, as the focus of the analysis and results is on equilibrium structures, discussion of kinetic processes might best be left out of this paper.

Even if the paper does not appear in Nat Commun, we would advise the authors to make these changes anyway.

Reviewer #2 (Remarks to the Author):

This manuscript introduces a new type of through-space interactions between oxygen lone pair atoms of carbonyl bonds and non-bonding pi orbitals of the carbonyl groups. This interaction is termed two sided $n \rightarrow \pi^*$ interactions. The authors synthesized a set of hydrazine molecules (see Fig. 1c), determined their X-ray structures and then performed electronic structure calculations. They analyzed the orbital energies using the NBO-method of Weinhold. Figs. 1e and 1f illustrate the favorable spatial contacts/overlaps between the molecular orbitals of the twisted hydrazine side-chains. The authors then searched the CSD and RCSB databases and discovered sizeable numbers of related interactions in the X-ray structures of small molecules (CSD) and of protein (RCSB). Fig. 4a shows that, in proteins, such contacts occur primarily in coils and turns which lack regular through-space interactions up to date. The authors suggest that this new type of interactions may provide a novel kind of stabilization to these elements, and may contribute to narrowing down the search space upon protein folding and thus to overcome the Levinthal's paradox.

Overall, the manuscript is well written and well accessible. The authors performed a good amount of experimental and theoretical work. The conclusions appear plausible and may have wide-ranging implications.

I only noticed one critical point that must be addressed plus a few minor points.

(1 – critical) The authors used X-ray coordinates of the small molecules as input for the electronic structure calculations. Although the resolution of the structures may be high (I could not find the resolution in Tables 1 and 2 nor in the text), it is common practice to relax molecular conformations during electronic structure calculations before providing final coordinates and energies. In the present case, with the twisted side chain conformations, it should be okay to keep the involved dihedral angles of the side chains frozen to their X-ray values.

However, the authors need to demonstrate – at least for their sequence of hydrazine derivatives in Fig. 1c – that geometry optimization of the remaining degrees of freedom (bond lengths, angles, and dihedrals) has a negligible effect on the overlap of the computed n/π^* orbitals and on the NBO energies in comparison to calculations done on the unrelaxed X-ray coordinates.

(2 – minor) p.6 two lines from bottom: insert how many molecules were randomly chosen from CSD

(3 – minor) p.7 seven lines from bottom and p.17 line 5: does „redundancy“ mean „pairwise sequence identity“? If yes, I suggest to use the latter expression.

(4 – minor) p.8 four lines from bottom: „in most number“ -> „in the largest number“

(5 – minor) p.10 lines 7/8: „providing an answer to Levinthal’s paradox“ should be replaced by something like „contribute to answering Levinthal’s paradox in addition to other concepts such as the free energy funnel model for protein folding (cite Onuchic, Wolynes, Proteins 1995)“.

Reviewer #3 (Remarks to the Author):

The manuscript reports the observation of previously uncharacterized reciprocating $n \rightarrow n^*$ interactions, wherein a carbonyl group accepting an interaction donates back to the carbonyl group from which it receives electron density. The existence of these reciprocal interactions was hypothesized based on the prediction that donation of an interaction into a particular carbonyl group should enhance the acceptor’s ability to donate to subsequent interactions. In support of this hypothesis, the authors provide crystal structures of several diacyl hydrazines that have structures consistent with reciprocal $n \rightarrow n^*$ interactions, which is further supported by quantum mechanical calculations. The authors then query the CSD and find numerous putative examples of reciprocal $n \rightarrow n^*$ interactions in diverse small molecules. Finally, the authors argue that similar interactions are common in proteins, based on geometries observed in protein crystal structures.

The characterization of protein structural motifs and their underlying determinants remains an important issue in protein science. As such, the identification of previously unrecognized interactions in proteins is likely to be of broad interest. Indeed, I am unaware of any reports explicitly addressing the possibility of such reciprocal $n \rightarrow n^*$ interactions. Though this hypothesis is very provocative, I believe that there are some important questions that need to be answered in order to be confident in the conclusions.

Overall, I find eight important points to be addressed:

1) The energy of these interactions in proteins is unclear. There are relatively few calculations performed on protein geometries, and those limited calculations sample a relatively small conformational space. In particular, the reported examples have angles of approach between 70° and 90° , whereas the data presented in Figure 3b suggest that many interactions have larger angles of approach. It is unclear, therefore, if all of the interactions identified herein have significant energy. Do they have energies similar to other $n \rightarrow n^*$ interactions? Or does the distorted geometry required for back donation affect the energy?

Because exhaustive calculations of $n \rightarrow n^*$ energies have been performed previously (see Bartlett et al. 2010), it should not be necessary to calculate energies for all of the observed geometries. Rather, calculations on a more representative subset should allow readers compare the energy of these newly identified interactions with the energy of previously reported $n \rightarrow n^*$ interactions. It would also be helpful if energy data for examples in proteins were incorporated into the main article, as opposed to residing in the supplement.

2) The most critical signature of the $n \rightarrow n^*$ interaction is the pyramidalization induced upon the acceptor carbonyl group, as the authors note in their introduction. What is the degree of pyramidalization in the synthetic diacyl hydrazines? Importantly, is pyramidalization observed in both carbonyl groups? This would lend strong credence to the existence of these interactions, and the data should already be available from the crystal structures. Pyramidalization data should also be available for CSD structures, though a certain amount of noise is to be expected (see Kamer, et al. 2013). Again, significant pyramidalization of both carbonyl groups participating in the interaction would greatly strengthen the authors’ claims.

3) The authors claim that d_2 distances in compounds 6-8 are shorter than d_1 because electronic

donation from the halogen lone pairs to the carbonyl n^* orbital make the carbonyl a stronger donor. However, no evidence is presented that such donation occurs. In fact, in all of the presented crystal structures, the halogen is proximal (cis) to the carbonyl oxygen, rather than the carbonyl carbon. Should halogen electron density be transferred to the carbonyl group, it should almost certainly occur through the carbonyl carbon, not oxygen. In addition, if such donation were to occur, it should be more apparent in 8 than in 6, due to the higher polarizability of bromine. It is therefore unlikely that the cause of d_2 constriction in compounds 6-8 is donation of electron density from the proximal halogen. The fact that d_2 is shorter than d_1 in these cases is likely due to other effects. Perhaps it is the case that, even in the absence of any back donation, enhancement of a single $n \rightarrow n^*$ interaction would decrease both distances.

4) I do not believe that the manuscript contains sufficient data to conclude that $n \rightarrow n^*$ interactions are cooperative. Though it is certainly intriguing that the energies of the paired $n \rightarrow n^*$ interactions are correlated, this does not necessarily imply cooperativity. As the authors note, higher $n \rightarrow n^*$ energies are observed when the donor-acceptor distance shortens. This should also cause a contraction of the second, reciprocal donor-acceptor distance, thereby increasing the strength of the second interaction independent of any polarization. It is therefore unclear if the increase in energy of the second interaction is due to polarization by the first interaction or if it merely results from the particular geometries observed. Though the authors are clear that the results are only suggestive of cooperativity, I feel that this point is too speculative given the data herein.

5) How much overlap is there between this dataset and previous examples? In particular, the authors report that 7% of residues form reciprocal $n \rightarrow n^*$ interactions; presumably, half of those residues were previously reported to engage in one-sided $n \rightarrow n^*$ interactions. However, it is difficult to compare this study with previous ones, as previous reports include an angular criterion for examining $n \rightarrow n^*$ interactions. How many of the interactions reported herein were not reported previously by Bartlett et al? That is, how many $n \rightarrow n^*$ interactions have we been missing by not considering reciprocal interactions?

6) The two rotatable bonds between each pair of carbonyl groups in the protein backbone both belong to the same amino acid residue, the second residue of the pair. Therefore, to plot the backbone dihedral angles of both residues in the Ramachandran plot in Figure 3d is very misleading, because the dihedral angles of the first residue do not affect the formation of the interaction. In fact, previous computations have shown no possibility for $n \rightarrow n^*$ interactions in many of the regions indicated in Figure 3d (see Bartlett, et al. 2010). It is unreasonable, therefore, to conclude that "two-sided $n \rightarrow n^*$ interactions have a widespread presence in the allowed regions of the Ramachandran plot." Plotting only the dihedral angles of the residue between the two relevant carbonyl groups should bring the data in line with expectations from previous computations.

7) With regard to the frequency of these interactions in different secondary structure types or different amino acids, it is important that the data be normalized. For example, in Figure 4a, could it be that the interactions are observed more frequently in "coil" than in "turn" because "coil" is simply more frequent than "turn"? The data from the last column of Extended figure 5a would be more appropriate. Similarly, the plots in Figures 4b and 4c should be normalized to the frequency of the amino acids. In particular, the observation that Leu-Pro is most common might simply be due to the fact that leucine is the single most common amino acid in proteins. When normalized to account for amino acid frequency, the data might show that other residues have stronger preferences for these reciprocal interactions. This is also relevant for Figure 4d, as hydrolases are the most well-represented class of enzymes in the PDB, and therefore might not be more prone to reciprocal $n \rightarrow n^*$ interactions than other classes. Finally, it is unclear what "frequency" is considered in Extended Figure 5b. Are there really proteins where 92% of residues engage in reciprocal $n \rightarrow n^*$ interactions? Perhaps "frequency" in this last case actually refers to the number of examples in that protein?

8) It is unclear where the data from Extended Figure 6 come from. Which structures were subjected to these calculations? Importantly, panel (a) appears to concern different molecules than panels (b/c), which are still different from those of panels (d/e). The datasets for these panels should be made clear. In addition, what is the horizontal axis for Extended Figures 6f/g? What conclusions are to be drawn from these data?

In addition to the important issues above, I believe that the manuscript would benefit from the following considerations.

1) How is the fit of Extended Figure 6a determined? What is the resulting model?

2) I think that it is important to note that many of the CSD hits, at least as described by Extended Figure 3c, are very conformationally constrained, either by additional rings, highly substituted centers, or stereoelectronic effects. This makes it all the more remarkable that reciprocal $n \rightarrow n^*$ interactions are observed in the much less-constrained protein backbone.

3) A ChemDraw figure would be helpful for interpreting Figure 2e/f.

4) The authors have not commented on the types of small molecules that form these reciprocal $n \rightarrow n^*$ interactions, which might reveal some interesting trends.

5) It would be helpful for the authors to show illustrative examples of the interactions they find in proteins, particularly in different secondary structure contexts. For example, the authors note that "α-helices that have two-sided $n \rightarrow n^*$ interactions are distorted, while the β-sheets having two sided $n \rightarrow n^*$ interactions are twisted." It would be very helpful for readers to be able to visualize these distortions, as they could have important consequences for protein structure. Such examples would, in my opinion, be more informative than the examples shown in Extended Figure 4.

Reviewer #1

Reviewer's comment: First, whilst I am in no doubt that the detail, quality and depth of this analysis and paper far outstrips any foregoing work, there are precedents for describing this type of arrangement of carbonyl groups in the literature, regardless of the mechanism of interaction. See:

Allen et al., Acta Cryst B54: 320-29 (1998) (Figure 1).

MacCallum et al (Milner-White and co), JMB (1995), 248:374-384 (Figure 3 and text).

For the background and context of this new work, these foregoing papers must be cited and discussed in a revised manuscript.

Our response: We thank the reviewer for pointing this out. We are aware that Allen and coworkers reported intermolecular anti-parallel $C=O \cdots C=O$ short contacts in solid state structures of 346 ketone dimers where the carbonyl oxygen of one ketone monomer makes an $O \cdots C$ short contact with the carbonyl carbon of the other ketone monomer to form two $O \cdots C$ short contacts that were shorter than 3.22 Å. As their work dealt with “intermolecular” $C=O \cdots C=O$ short contacts that were dipolar in nature, we did not discuss them with our “intramolecular” $C=O \cdots C=O$ $n \rightarrow \pi^*$ interactions. As suggested by the reviewer, for the background and context of our new work, now we have discussed this work in the revised manuscript.

Maccallum et al reported a similar arrangements of carbonyl groups in right-twisted β -strands and observed two chemically distinct dipolar $C=O \cdots C=O$ short contacts³². However, these $C=O \cdots C=O$ short contacts were considerably longer than the sum of van der Waals radii of C and O atoms. This work is particularly relevant to our discussion and we have now discussed this work in our revised manuscript.

Reviewer's comment: Page 4. I am confused by the energies of interactions 1 and 2. These are in effect closed systems, and one might anticipate that the electrons become evenly distributed. So why the difference in energy between the "1st" and "2nd" $n \rightarrow \pi^*$ interaction? I'm not sure that I buy the explanation given. Moreover, the argument made does not tally with the data given in Table 2, where all possibilities are seen, interaction 1 > interaction 2, 2 > 1 and 1 \approx 2.

Our response: Compounds **1-8** are N, N'-diacylhydrazine-based compounds. The C=O...C=O interaction energies in **1-8** depend on the substituents near the carbonyl groups and there is a correlation between the two interactions. For example, let us compare molecules **3** and **4** where Y is same (Y = CH₃) but X is different (**3**, X = CH₃; **4**, X=CH₂Cl) (See the Figure below, **RRC Fig. 1**). It is known that the lone pairs of α -halo groups can donate electrons into the antibonding orbital (π^*) of the adjacent carbonyl group (C=O). We observed that in compound **4**, electron donation occurs to both σ^* and π^* orbitals of the C=O bonds of CO-II from the lone pairs of Cl atom positioned at the α -position (**RRC Table 1** below). Such electron donation enhances the ability of the carbonyl group II (CO-II) to partake in the $n \rightarrow \pi^*$ interaction. Therefore, CO-II in **4** is a better electron donor than CO-II group in **3**, which is reflected in the shorter d_2 in **4** ($d_2 = 3.039 \text{ \AA}$) than **3** ($d_2 = 3.365 \text{ \AA}$). This is also evident from the higher second order perturbation energy [$E^2_{(n \rightarrow \pi^*)}$] for donation of electrons from CO-II to CO-I in **4** [$E^2_{(n \rightarrow \pi^*)} = 0.34 \text{ kcal.mol}^{-1}$] compared to **3** [$E^2_{(n \rightarrow \pi^*)} = 0.02 \text{ kcal.mol}^{-1}$]. Due to this electron donation from CO-II to CO-I, the CO-I group in **4** will be more polarized than the CO-I group in **3** and the carbonyl oxygen of CO-I in **4** will become better donor and donation from CO-I to CO-II will be stronger in **4** than **3**. Accordingly, we observed a decrease in d_1 and increase in $E^1_{(n \rightarrow \pi^*)}$ in **4** [$d_1 = 3.103 \text{ \AA}$ and $E^1_{(n \rightarrow \pi^*)} = 0.10 \text{ kcal.mol}^{-1}$] compared to **3** [$d_1 = 3.367 \text{ \AA}$ and $E^1_{(n \rightarrow \pi^*)} = 0.01 \text{ kcal.mol}^{-1}$]. Therefore, the 1st and the 2nd interaction energies in these compounds are different. We have provided the donation of electrons from halogens to CO-II in compounds **4, 6-8** in **RRC Table 1** below. These results are included in Supplementary Table 4 in the revised manuscript.

RRC Fig. 1. Chemical structures of compounds **3** and **4** showing the positions of the carbonyl groups CO-I and CO-II and the possibility of Cl to CO-II electron donation.

RRC Table 1. Electron donation from the halogen (Cl or Br) lone pairs to the antibonding π^* and σ^* orbitals of nearby carbonyl C=O bonds in compounds **4**, **6-8**. The calculations are carried out at B3LYP/6-311+G(2d,p) level of theory. X-C-C=O torsion angles are from crystal geometries.

comp	X-C-C=O torsion angle (X = Cl or Br)	$E_{(n \rightarrow \pi^*)}$ kcal.mol ⁻¹	$E_{(n \rightarrow \sigma^*)}$ kcal.mol ⁻¹	$E^t = E_{(n \rightarrow \pi^*)} + E_{(n \rightarrow \sigma^*)}$ kcal.mol ⁻¹
4	-1.2	0.23	0.44	0.67
6	33.3	0.26	0.43	0.69
7	-61.6	1.06	0.30	1.36
8	-35.8	0.19	0.44	0.63

Overall, the strength of $n \rightarrow \pi^*$ interactions depends on the O...C distance; shorter is the O...C distance stronger is the interaction. The O...C distance can be tuned by using electron donating and withdrawing substituents near the carbonyl groups as shown for compounds **1-8** (Table 1 in the revised manuscript). We have now included the contribution of donation of electrons from the α -halo groups into the antibonding orbital (π^*) of the adjacent carbonyl group (C=O) in the revised manuscript to explain these observations.

In Table 2, we have discussed the $n \rightarrow \pi^*$ interactions in molecules obtained from the CSD search. These molecules have diverse structures. Unlike compounds **1-8** that are all N, N-diacylhydrazine-based molecules, the molecules in Table 2 are not based on a particular structural type. Therefore, we cannot expect a correlation in the reciprocal $n \rightarrow \pi^*$ interactions of the molecules.

Moreover, we have randomly chosen the two carbonyl groups as CO-I and CO-II in these molecules in Table 2. Only correlation we can expect here is an increase in the interaction energy [$E^1_{(n\rightarrow\pi^*)}$ or $E^2_{(n\rightarrow\pi^*)}$] with a decrease in O...C distance (d_1 or d_2).

Reviewer's comment: Figures 2 and 3. The distances d_1 and d_2 are described and shown for the systems that test positive for the two-sided $n\rightarrow\pi^*$ only. But what is the distribution of these distances in unconstrained systems. (I realise that this probably only has meaning in the context of $i, i+1$ residues in proteins.) This is important to see if the short distances are really outliers in the full distributions or not.

Our Response: We agree with the reviewer that the distribution of unconstrained distances d_1 and d_2 can give us information regarding whether the short distances are outliers in the full distributions or not. As pointed out by the reviewer, such a distribution would be meaningful for proteins as the carbonyl groups of i^{th} and $(i+1)^{\text{th}}$ residues in proteins are always separated by three covalent bonds. For comparison, we have now plotted the distribution of distances d_1 and d_2 for all amino acid residues in the 2184 proteins studied in this work (see **RRC Fig. 2** below). To find out if the shorter distances (d_1 and $d_2 \leq 3.2 \text{ \AA}$) are outliers in the overall distribution, we used two different methods; two standard deviation (SD) method and outlier labeling rule method [Hoaglin, D. C. & Iglewicz, B. Fine tuning some resistant rules for outlier labeling, *J. Am. Stat. Assoc.* **82**, 1147-1149 (1987)]. Both methods suggest that short distances d_1 & $d_2 \leq 3.0 \text{ \AA}$ are not completely outliers in the overall distribution. They may be considered as tail of short distances in the full distribution.

RRC Fig. 2. Plot showing the distribution of O...C distances (d_1 and d_2) in all amino acids in 2269 proteins studied. The red box with the red point at (3.2, 3.2) indicates reciprocal interactions (d_1 and $d_2 \leq 3.2 \text{ \AA}$). 2 Standard deviation (SD) analyses are presented by green lines. d_1 and d_2 values lower than $(\mu - 2\sigma)$ and higher than $(\mu + 2\sigma)$ represents outliers [μ =mean; σ = standard deviation). Outlier labeling rule analyses using 2.2 as multiplier are presented by blue lines. M is the median and d_1 and d_2 values lower than L and higher than U represents outliers.

Reviewer's comment: None of the data plotted in Figs. 4a-c are normalized, ie raw counts or frequencies are given rather than percentages, or, better, propensities. This is also important. It is best illustrated by example: Clearly, in panel a two-sided $n \rightarrow \pi^*$ are rare in alpha and beta structures, but we can't compare these numbers with coil and turn, because these states occur in different amounts

across the whole PDB. Therefore, the numbers for each state need to be normalized somehow for the % of that state in the PDB as a whole. Numbers could then be compared. This will help the authors' argument I'm sure, as coil is rarer than alpha etc.

The same is true for the amino acids in panel b. Met, Cys and Trp are the amino acids that occur least frequently in proteins, so it is no surprise that they sit on the RHS of the plot. I suspect that normalising for amino acid occurrence in the PDB would flatten all amino acids other than Pro and Gly, which would add weight to the authors' argument.

While on this, panel c seems redundant as it only highlights the Pro and Gly-containing pairs really. Panel d is hard to fathom and explain, and should be removed in my view.

Our Response: We agree with the reviewer that normalized data/plots would provide better insights. We have now provided normalized data/plots for secondary structure distribution, amino acid distribution and amino acid pair distribution instead of absolute numbers.

1. **Secondary structure (SS) distribution plot:** Percentage of amino acids involved in reciprocal interactions is defined as:

(Number of amino acids in the SS involved in reciprocal $n \rightarrow \pi^*$ interaction x 100) ÷ (Total number of amino acids in that SS)

RRC Fig. 3. Plot of percentage of amino acids involved in reciprocal interactions in various secondary structures.

2. To find out % distribution of each amino acid to participate in reciprocal $n \rightarrow \pi^*$ interaction, we used the following formula

(Number of the particular amino acid in reciprocal $n \rightarrow \pi^*$ interaction x 100) ÷ (Total number of that particular amino acid in the proteins studied here)

RRC Fig. 4. Plot showing percentage distribution of amino acids involved in reciprocal $C=O \cdots C=O$ interactions.

3. To find out the % distribution of amino acid pairs to participate in reciprocal $n \rightarrow \pi^*$ interactions, we used the following formula.

(Number of a particular amino acid pair in the reciprocal $n \rightarrow \pi^*$ interaction x 100) ÷ (Total number of that amino acid pair in the proteins studied here)

RRC Fig. 5. Plot showing percentage distribution of amino acid pairs involved in reciprocal $C=O \cdots C=O$ interactions.

The changes that have occurred to our results due to normalization is discussed below.

1. In case of the abundance of reciprocal interactions in secondary structures, the trend is almost similar before and after normalization except that reciprocal interactions are found more in 3_{10} -helices compared to α -helices. α -helices are predominant in most proteins but only a handful of α -helix residues are involved in reciprocal interaction. The abundance of reciprocal interactions in various secondary structures is now presented in Table 4 of the revised manuscript.
2. After normalization, the amino acid percentage distribution differs from the frequency distribution provided earlier although proline remains the most abundant amino acid to

participate in reciprocal $n \rightarrow \pi^*$ interaction. However, normalization has increased the percentage distribution of proline residues (24%). These results are presented in Figure 4c of the revised manuscript.

3. The plot of the amino acid pair that participate in reciprocal $n \rightarrow \pi^*$ interactions after normalization has changed significantly from the one before normalization. After normalization, the Pro-Pro residue pair is the most predominant one to partake in the reciprocal $n \rightarrow \pi^*$ interaction. These results are presented in Figure 4d of the revised manuscript.

We have now incorporated these changes in the revised manuscript. As suggested by reviewer, we have also removed plot 4d.

Reviewer's comment: While on Pro, have the authors considered cis/trans forms and how these differ in propensity to form two-sided $n \rightarrow \pi^*$ interactions?

Our Response: It is known that $n \rightarrow \pi^*$ interaction is possible only if the proline residue exists in *trans* conformation. Based on this observation, the ratio of isomers ($K_{\text{trans/cis}}$) is used to report the energy of $n \rightarrow \pi^*$ interaction using NMR spectroscopy [*J. Am. Chem. Soc.* **135**, 7843-7846 (2013)]. Therefore, we assumed that the conformation of Pro amide bonds involved in reciprocal interactions would also be *trans*. However, after this query from the reviewer, we looked into the conformation of the amide bond (C-N-C=O dihedral angle) in Pro in Leu-Pro and Lys-Pro amino acid pairs (two among the most abundant amino acid pairs to participate in reciprocal interactions) and observed that the Pro conformations in the residues involved in reciprocal interactions are *trans*. Please see the files ROM-1-Leu-Pro and ROM-2-Lys-Pro provided as review only material.

Reviewer's comment: *I mentioned my preference for “reciprocated $n \rightarrow \pi^*$ interactions” as I find “two-sided $n \rightarrow \pi^*$ interactions” confusing. The authors may wish to consider this alternative name.

Our response: We thank the reviewer for this suggestion. We have now modified the name of the interaction to “reciprocal interaction” and title of the manuscript is now changed to “Reciprocal carbonyl-carbonyl interactions in small molecules and proteins.”

Reviewer's comment: Finally, I am not taken by the arguments that these could influence folding. Even at $\frac{1}{4}$ kcal/mol for each $n \rightarrow \pi^*$ interaction these would perturb equilibria between states only slightly. This is before solvent is considered. Surely, water competes well for solvation of carbonyl groups in a flexible unfolded state? I would recommend that this section is removed or trimmed at least. In other words, as the focus of the analysis and results is on equilibrium structures, discussion of kinetic processes might best be left out of this paper.

Our response: We agree with the reviewer that mixing kinetics and thermodynamic is not a good idea. However, as these interactions are most abundant in polyproline II helices that are the most abundant secondary structure in unfolded proteins, gives rise to the possibility of these interactions playing some role in protein folding. Also, turn regions that are stabilized by reciprocal interactions are known to act as nucleation sites for protein folding. Therefore, an open question is how important such reciprocal interactions might be for protein folding. We have now modified the discussion regarding protein folding both in the revised manuscript and the abstract.

Reviewer #2

Reviewer's comment: The authors used X-ray coordinates of the small molecules as input for the electronic structure calculations. Although the resolution of the structures may be high (I could not find the resolution in Tables 1 and 2 nor in the text), it is common practice to relax molecular conformations during electronic structure calculations before providing final coordinates and energies. In the present case, with the twisted side chain conformations, it should be okay to keep the involved dihedral angles of the side chains frozen to their X-ray values.

However, the authors need to demonstrate – at least for their sequence of hydrazine derivatives in Fig. 1c – that geometry optimization of the remaining degrees of freedom (bond lengths, angles, and dihedrals) has a negligible effect on the overlap of the computed n/π^* orbitals and on the NBO energies in comparison to calculations done on the unrelaxed X-ray coordinates.

Our response: The resolution of the X-ray crystal structures of the compound **1-8** were high (See **RRC Table 1** below). We have now included this information in the Supplementary Table 2 of the revised manuscript. We agree with the reviewer that providing optimized geometry and electronic energy calculation on optimized geometry is the common practice. However, as we have discussed in the manuscript, the donor and acceptor abilities of the carbonyl groups in compounds **1-8** are dependent on the substituents attached to the carbonyl groups and their geometrical arrangements. For example, the halogen atoms present in the substituent X in compounds **4, 6-8** highly influence the donor ability of the nearby carbonyl group (CO-II) (See **RRC Fig. 6** below). A change in the orientation of the halogen with respect to the CO-II group would change the donor ability of the CO-II group. We carried out optimization of compound **6** by freezing the coordinates of the atoms shown within the box (see the RHS ball and stick **RRC Fig. 7b** below). However, during optimization the chlorine (Cl) atom moved to an anti-periplanar geometry (*trans*) from an almost syn-periplanar

geometry (*cis*) with respect to the oxygen atom of the nearby carbonyl group (CO-II). In fact, when a completely relaxed geometry optimization of **6** was done at B3LYP/6-311+G(2d,p) level of theory, the Cl atom moved to an anti-periplanar position and the geometry of **6** became almost planar without any reciprocal interaction. In such a scenario, one would have to come up with the best theoretical method and basis set to study these interaction, which itself would be a separate project for investigation. As this study deals with the possibility of reciprocal interactions in the crystal geometries of small molecules and proteins, to avoid such deviation from crystal geometries, we avoided free optimization the molecules.

RRC Table 2. Resolution of crystal structures of compounds **1-8**.

Comp	Resolution (Å)	Comp	Resolution (Å)	Comp	Resolution (Å)	Comp	Resolution (Å)
1	0.82	3	0.80	5	0.82	7	0.80
2	0.84	4	0.82	6	0.82	8	0.82

RRC Fig. 6. a, Chemical structure of compounds **4, 6-8. b**, Chemical structure of compound **6** showing the positions of carbonyl groups CO-I and CO-II and possibility of electron donation from Cl to CO-II.

RRC Fig. 7. a, Crystal geometry of compound **6**. **b**, B3LYP/6-311+G(2d,p) level optimized geometry of **6** obtained by freezing the coordinates of the atoms shown within the red box.

Reviewer's comment: p.6 two lines from bottom: insert how many molecules were randomly chosen from CSD.

Our response: We have now mentioned the number of molecules (30) that were randomly chosen from the CSD in the revised manuscript.

Reviewer's comment: p.7 seven lines from bottom and p.17 line 5: does “redundancy” mean “pairwise sequence identity”? If yes, I suggest to use the latter expression.

Our response: We have used 10 % redundancy for our PDB search which means that a protein from PDB will not be included in the search if it's sequence similarity more than 90% to any of the protein present in the PDB. Both redundancy and pairwise sequence identity are commonly used [see Nat. Chem. Biol. **6**, 615-620 (2010)]. We have used both the terminologies in the manuscript now.

Reviewer's comment: p.8 four lines from bottom: „in most number“ -> „in the largest number“

Our response: most number is now changed to the largest number in the revised manuscript.

Reviewer's comment: p.10 lines 7/8: “providing an answer to Levinthal’s paradox” should be replaced by something like “contribute to answering Levinthal’s paradox in addition to other concepts such as the free energy funnel model for protein folding (cite Onuchic, Wolynes, Proteins 1995)”.

Our response: This section is now modified. Please see the revised manuscript.

Reviewer #3

Reviewer's comment:

The energy of these interactions in proteins is unclear. There are relatively few calculations performed on protein geometries, and those limited calculations sample a relatively small conformational space. In particular, the reported examples have angles of approach between 70° and 90°, whereas the data presented in Figure 3b suggest that many interactions have larger angles of approach. It is unclear, therefore, if all of the interactions identified herein have significant energy. Do they have energies similar to other $n \rightarrow \pi^*$ interactions? Or does the distorted geometry required for back donation affect the energy?

Because exhaustive calculations of $n \rightarrow \pi^*$ energies have been performed previously (see Bartlett et al. 2010), it should not be necessary to calculate energies for all of the observed geometries. Rather, calculations on a more representative subset should allow readers compare the energy of these newly identified interactions with the energy of previously reported $n \rightarrow \pi^*$ interactions. It would also be helpful if energy data for examples in proteins were incorporated into the main article, as opposed to residing in the supplement.

Our response: The strength of $n \rightarrow \pi^*$ interactions depends on the O...C distance. As the O...C distance increases, the stabilization due individual $n \rightarrow \pi^*$ interactions decreases. This can be seen from Figure 2c-2d of the revised manuscript for the molecules obtained from the CSD search where the distances d_1 and d_2 increase as the angles θ_1 and θ_2 deviate from the Bürgi-Dunitz trajectory.

To confirm that reciprocal $n \rightarrow \pi^*$ interaction is present when the angle of approach is greater than 90°, we have now performed additional NBO calculations on representative subsets that cover the complete range of observed O...C=O angles (70-110°) and observed that reciprocal $n \rightarrow \pi^*$ interaction is present even the angle of approach is greater than 90 °. The calculations were done for

amino acid pairs extracted from the PDB as well as small molecules obtained from the CSD that have reciprocal interactions. We observed substantial $\text{C}=\text{O}\cdots\text{C}=\text{O}$ $\pi\rightarrow\pi^*$ interactions in molecules having θ_1 and θ_2 values $> 90^\circ$ (Supplementary Table 6 and Supplementary Table 8) for both CSD molecules and amino acid pairs taken from PDB. In some cases, $\pi\rightarrow\pi^*$ interactions are even stronger than $n\rightarrow\pi^*$ interactions. When θ_1 and θ_2 values were $< 90^\circ$, $\pi\rightarrow\pi^*$ interactions were observed for molecules having relatively stronger $n\rightarrow\pi^*$ interactions (both d_1 and $d_2 < 2.90 \text{ \AA}$). This indicates that although the individual $n\rightarrow\pi^*$ interaction in reciprocal interaction is weak, the sum of the two $n\rightarrow\pi^*$ interactions together with two $\pi\rightarrow\pi^*$ interactions provide substantial stabilization to small molecule and proteins. Based on NBO calculations at B3LYP/6-311+G(2d,p) level, we observed 0.11-3.37 kcal.mol^{-1} (with an average $0.98 \text{ kcal.mol}^{-1}$) stabilization of small molecules from reciprocal $\text{C}=\text{O}\cdots\text{C}=\text{O}$ interactions (See the last column in Supplementary Table 6). Similarly, we observed that reciprocal $\text{C}=\text{O}\cdots\text{C}=\text{O}$ interactions contribute 0.27-4.41 kcal.mol^{-1} (with an average value $1.34 \text{ kcal.mol}^{-1}$) stabilization to proteins per each amino acid pair (See the last column in Supplementary Table 8). These results are now discussed in the revised manuscript (See Table 2 and Table 3) and also incorporated in the Supplementary Table 6 and Supplementary Table 8.

Reviewer's comment: The most critical signature of the $n\rightarrow\pi^*$ interaction is the pyramidalization induced upon the acceptor carbonyl group, as the authors note in their introduction. What is the degree of pyramidalization in the synthetic diacyl hydrazines? Importantly, is pyramidalization observed in both carbonyl groups? This would lend strong credence to the existence of these interactions, and the data should already be available from the crystal structures. Pyramidalization data should also be available for CSD structures, though a certain amount of noise is to be expected (see Kamer, et al. 2013). Again, significant pyramidalization of both carbonyl groups participating in the interaction would greatly strengthen the authors' claims.

Our response: We agree that the pyramidity of the acceptor carbonyl carbon atom measured by parameters Δ and Θ is another important signature of $n \rightarrow \pi^*$ interactions. Positive values of Δ and Θ indicate pyramidalization of the acceptor carbonyl carbon towards the donor oxygen atom whereas negative values of Δ and Θ indicate pyramidalization of the acceptor carbon away from the donor oxygen atom. We have now tabulated the Δ and Θ values of compounds **1-8** in the Table 1 of the revised manuscript. In compounds **1-8**, however, we have not observed a correlation of pyramidity (Θ) with $O \cdots C$ distance and the strength of $n \rightarrow \pi^*$ interactions. One reason for this could be the stronger donation from the α -halogen atoms to the nearby carbonyl, which would force the acceptor carbonyl carbons towards the halogens atoms away from the donor oxygen atoms. Also, the crystal packing forces may have some influence in the observed geometries and the pyramidalization of the two nitrogen atoms between the carbonyl groups may influence the pyramidalization of the acceptor carbonyl carbons. Moreover, the individual $n \rightarrow \pi^*$ interactions in compounds **1-8** may not be strong enough to exert a significant effect on pyramidalization of the carbonyl carbons.

We observed positive values of Δ and Θ for the carbonyl carbons in most of the molecules from the CSD listed in Table 2, which indicate their pyramidalization towards the donor oxygen atoms. The plots of Θ with $O \cdots C$ distances and the strength of the reciprocal interactions in compounds obtained from the CSD search are shown in Supplementary Fig. 5 (See **RRC Fig. 8 below**). Although the correlation between pyramidity of second carbonyl (CO-II) carbon (Θ_2) and d_1 looks better than the correlation between pyramidity of first carbonyl (CO-I) carbon (Θ_1) and d_2 , the CO-I and CO-II are chosen completely randomly in these molecules. As the pyramidalization also depends on other factors like θ and the elasticity of the carbonyl group, a strong correlation between pyramidalization and the $O \cdots C$ distance and strength of $n \rightarrow \pi^*$ interactions may not be observed in these molecules having different types of carbonyl groups as well as different θ values.

In proteins also, we observed positive pyramidity (Θ) of both the acceptor carbonyl carbon atoms involved in reciprocal interaction, which indicates their pyramidalization towards the donor oxygen atoms (Table 3). The plots of Θ with $O\cdots C$ distances and the strength of the reciprocal interactions in the amino acid pairs involved in the reciprocal interactions are shown in Supplementary Figure 7 (See **RRC Fig. 9** below).

RRC Fig. 8. Plots of pyramidity (Θ) of acceptor carbonyl carbon atoms with $O\cdots C$ (d_1 and d_2) distances and $C=O\cdots C=O$ interaction energies in the molecules obtained from the CSD search shown in Table 2. **a**, Θ_2 vs. d_1 . **b**, Θ_1 vs. d_2 . **c**, Θ_2 vs. $E^1_{(n\rightarrow\pi^*)}$. **d**, Θ_1 vs. $E^2_{(n\rightarrow\pi^*)}$. **e**, Θ_2 vs. $[E^1_{(n\rightarrow\pi^*)} + E^1_{(\pi\rightarrow\pi^*)}]$. **f**, Θ_1 vs. $[E^2_{(n\rightarrow\pi^*)} + E^2_{(\pi\rightarrow\pi^*)}]$. d_1 , d_2 , Θ_1 , Θ_2 , $E^1_{(n\rightarrow\pi^*)}$ and $E^2_{(n\rightarrow\pi^*)}$ values are taken from Table 2. $E^1_{(\pi\rightarrow\pi^*)}$ and $E^2_{(\pi\rightarrow\pi^*)}$ values are taken from Supplementary Table 6.

RRC Fig. 9. Plots of pyramidity (Θ) of acceptor carbonyl carbon atoms with O...C (d_1 and d_2) distances and C=O...C=O interaction energies in the amino acid pairs shown in Table 3. **a**, Θ_2 vs. d_1 . **b**, Θ_1 vs. d_2 . **c**, Θ_2 vs. $E^1_{(n \rightarrow \pi^*)}$. **d**, Θ_1 vs. $E^2_{(n \rightarrow \pi^*)}$. **e**, Θ_2 vs. $[E^1_{(n \rightarrow \pi^*)} + E^1_{(\pi \rightarrow \pi^*)}]$. **f**, Θ_1 vs. $[E^2_{(n \rightarrow \pi^*)} + E^2_{(\pi \rightarrow \pi^*)}]$. d_1 , d_2 , Θ_1 , Θ_2 , $E^1_{(n \rightarrow \pi^*)}$ and $E^2_{(n \rightarrow \pi^*)}$ values are taken from Table 3. $E^1_{(\pi \rightarrow \pi^*)}$ and $E^2_{(\pi \rightarrow \pi^*)}$ values are taken from Supplementary Table 8.

Reviewer's comment: The authors claim that d_2 distances in compounds **6-8** are shorter than d_1 because electronic donation from the halogen lone pairs to the carbonyl π^* orbital make the carbonyl a stronger donor. However, no evidence is presented that such donation occurs. In fact, in all of the presented crystal structures, the halogen is proximal (*cis*) to the carbonyl oxygen, rather than the carbonyl carbon. Should halogen electron density be transferred to the carbonyl group, it should almost certainly occur through the carbonyl carbon, not oxygen. In addition, if such donation were to occur, it should be more apparent in **8** than in **6**, due to the higher polarizability of bromine. It is therefore unlikely that the cause of d_2 constriction in compounds **6-8** is donation of electron density from the proximal halogen. The fact that d_2 is shorter than d_1 in these cases is likely due to other effects. Perhaps it is the case that, even in the absence of any back donation, enhancement of a single $n \rightarrow \pi^*$ interaction would decrease both distances.

Our response: We have now provided evidence for electron donation from the α -halogen lone pairs to carbonyl group's σ^* and π^* orbitals (see **RRC Table 3** below). These results are included in supplementary Table 4 in the revised manuscript. We agree with the reviewer that a halogen atom anti-periplanar (*trans*) to the carbonyl oxygen would donate electrons much better to the carbonyl group than when it is in syn-periplanar (*cis*). As can be seen from the X-C-C=O torsion angles in the **RRC Table 3** below, the halogen atoms in the crystal geometries are not exactly *cis* to the carbonyl group except in compound **4**. There is substantial electron donation from the halogen atoms to the σ^* and π^* orbitals of the C=O group, which should increase the donor ability of the carbonyl group CO-II.

As can be seen from the Table below, **6** and **8** have similar X-C-C=O torsion angles but stabilization due to donation from Br lone pairs to CO-II in **8** is less than the stabilization due to donation from Cl lone pairs to CO-II in **6** even though Br is more polarizable. This expected as the

size of Br orbitals will be much higher than C and O orbitals that will hamper efficient delocalization between the Br lone pairs and the σ^* and π^* orbitals of C=O bond. Compared to Br, relatively smaller Cl lone pair orbitals will have better overlap with σ^* and π^* orbitals of the C=O group. These observations are now included in the revised paper and Supplementary Table 4. The chemical structures of compounds **4**, **6-8** are shown in **RRC Fig. 10** below.

RRC Fig. 10. Chemical structures of compounds **4**, **6-8**.

RRC Table 3. Electron donation from the halogen (Cl or Br) lone pairs to the antibonding π^* and σ^* orbitals of nearby carbonyl C=O bonds in compounds **4**, **6-8**. The calculations are carried out at B3LYP/6-311+G(2d,p) level of theory. X-C-C=O torsion angles are from crystal geometries.

comp	X-C-C=O torsion angle (X = Cl or Br)	$E_{(n \rightarrow \pi^*)}$	$E_{(n \rightarrow \sigma^*)}$	$E^t = E_{(n \rightarrow \pi^*)} + E_{(n \rightarrow \sigma^*)}$
		kcal.mol ⁻¹	kcal.mol ⁻¹	kcal.mol ⁻¹
4	-1.2	0.23	0.44	0.67
6	33.3	0.26	0.43	0.69
7	-61.6	1.06	0.30	1.36
8	-35.8	0.19	0.44	0.63

Reviewer's comment: I do not believe that the manuscript contains sufficient data to conclude that $n \rightarrow \pi^*$ interactions are cooperative. Though it is certainly intriguing that the energies of the paired $n \rightarrow \pi^*$ interactions are correlated, this does not necessarily imply cooperativity. As the authors note, higher $n \rightarrow \pi^*$ energies are observed when the donor–acceptor distance shortens. This should also cause a contraction of the second, reciprocal donor–acceptor distance, thereby increasing the strength of the second interaction independent of any polarization. It is therefore unclear if the increase in

energy of the second interaction is due to polarization by the first interaction or if it merely results from the particular geometries observed. Though the authors are clear that the results are only suggestive of cooperativity, I feel that this point is too speculative given the data herein.

Our response: We believe that the reciprocal $n \rightarrow \pi^*$ interactions between the carbonyl groups in compounds **1-8** are cooperative. Our data suggest that an increase in $n \rightarrow \pi^*$ interaction from one side also leads to an increase in the $n \rightarrow \pi^*$ interaction from the other side in compounds **1-8** (**RRC Fig. 11a-b below**). The cooperativity of reciprocal $n \rightarrow \pi^*$ interactions in **1-8** becomes clear when we compare the interactions in compound pairs (**3, 4**) and (**5, 6**). In compound **4**, as discussed above, the presence of the Cl atom increases electron donation from CO-II to CO-I. This induces much higher back donation of electron from CO-I to CO-II in **4** than **3** even though both **3** and **4** have the same substituent (CH₃) attached to the CO-I. Similarly, if we compare compounds **5** and **6**, the presence of Cl in **6** increases electron donation from CO-II to CO-I compared to **5**. This in turn increases back donation from CO-I to CO-II in **6** compared to **5** although both **5** and **6** have the same substituent (OCH₃) attached to the CO-I. These results indicate that the reciprocal interactions in these N, N-diacylhydrazines (**1-8**) could be cooperative. We observed that shortening of one donor-acceptor distance does not always lead to a contraction of the second, reciprocal donor-acceptor distance. For example, if we look at Figures 2a and 3a discussed in the revised manuscript, the d_1 vs. d_2 plots are scattered and, in most cases, d_1 does not increase or decrease with an increase or decrease of d_2 , which suggests that shortening of one donor-acceptor distance does not necessarily lead to a contraction of the second, reciprocal donor-acceptor distance. Therefore, the correlation observed between the reciprocal $n \rightarrow \pi^*$ interactions in compounds **1-8** cannot be generalized. However, to ascertain whether the origin of this correlation between the reciprocal $n \rightarrow \pi^*$ interactions in **1-8** is a result of cooperative

effects or merely a result of the particular geometries adopted by **1-8** needs further investigation. The plots shown below (**RRC Fig.11**) are now included in Fig. 1g-h of the revised manuscript.

RRC Fig. 11. a, Plot showing correlation between O...C distances (d_1 and d_2) in compounds **1-8** [Linear fitting; Pearson correlation coefficient = 0.9906]. **b**, Plot showing correlation between reciprocal $n \rightarrow \pi^*$ interaction energies [$E^1_{(n \rightarrow \pi^*)}$ and $E^2_{(n \rightarrow \pi^*)}$] in compounds **1-8** [Linear fitting; Pearson correlation coefficient = 0.938].

Reviewer's comment: How much overlap is there between this dataset and previous examples? In particular, the authors report that 7% of residues form reciprocal $n \rightarrow \pi^*$ interactions; presumably, half of those residues were previously reported to engage in one-sided $n \rightarrow \pi^*$ interactions. However, it is difficult to compare this study with previous ones, as previous reports include an angular criterion for examining $n \rightarrow \pi^*$ interactions. How many of the interactions reported herein were not reported previously by Bartlett et al? That is, how many $n \rightarrow \pi^*$ interactions have we been missing by not considering reciprocal interactions?

Our response: In a previous study, Bartlett et al reported one-sided $n \rightarrow \pi^*$ interactions with $d \leq 3.20$ Å and $99^\circ \leq \theta \leq 119^\circ$ [Nat. Chem. Biol. **6**, 615-620 (2010)]. As we have applied the same distance (d

$\leq 3.20 \text{ \AA}$) and resolution ($< 1.6 \text{ \AA}$) criteria, the reciprocal interactions observed here for angles $99^\circ \leq \theta_1, \theta_2 \leq 119^\circ$ must have been reported by Bartlett et al as one-sided $n \rightarrow \pi^*$ interactions. However, our data (downloaded on 19th January 2016) contains many additional proteins apart from what Bartlett et al studied (downloaded on 18th December 2007). Also, we have used a lower redundancy (pairwise sequence identity) value (10%) than Bartlett et al (30%), which will also increase the number of proteins in our data. As can be seen from **RRC Fig. 12** below, the distribution of θ_1 and θ_2 in the range of 99° - 119° (regions II, III and IV) is a very small percentage (6.5%) of the total number of reciprocal $\text{C}=\text{O} \cdots \text{C}=\text{O}$ interactions that are being reported here. In fact, the overlap of reciprocal interactions with previously reported $n \rightarrow \pi^*$ interactions would be lower than 6.5% as Figure 3b contains many additional proteins that were not there in the data set of Bartlett et al. Therefore, at least 93.5% of the reciprocal $\text{C}=\text{O} \cdots \text{C}=\text{O}$ interactions (region I) reported here are novel. These results are now explained in the revised manuscript and included in Figure 3b of the revised manuscript.

RRC Table 4. Data used to calculate % of overlap between our and previously reported $n \rightarrow \pi^*$ interactions.

Total number of reciprocal $n \rightarrow \pi^*$ interactions found in PDB search = 19142

Total number of reciprocal $n \rightarrow \pi^*$ interactions that come under = 1257

criteria of $n \rightarrow \pi^*$ interaction reported by Bartlett *et al*

RRC Fig. 12. Plot showing the distribution of $\angle\text{O}\cdots\text{C}=\text{O}$ angles θ_1 and θ_2 in amino acid pairs in proteins having reciprocal $\text{C}=\text{O}\cdots\text{C}=\text{O}$ interactions. Region I represents newly discovered $n\rightarrow\pi^*$ interactions and regions II, II and IV represent $n\rightarrow\pi^*$ interactions reported by Bartlett et al previously.

Reviewer's comment: The two rotatable bonds between each pair of carbonyl groups in the protein backbone both belong to the same amino acid residue, the second residue of the pair. Therefore, to plot the backbone dihedral angles of both residues in the Ramachandran plot in Figure 3d is very misleading, because the dihedral angles of the first residue do not affect the formation of the interaction. In fact, previous computations have shown no possibility for $n\rightarrow\pi^*$ interactions in many of the regions indicated in Figure 3d (see Bartlett, et al. 2010). It is unreasonable, therefore, to conclude that “two-sided $n\rightarrow\pi^*$ interactions have a widespread presence in the allowed regions of the Ramachandran plot.” Plotting only the dihedral angles of the residue between the two relevant carbonyl groups should bring the data in line with expectations from previous computations.

Our response: We thank the reviewer for pointing this out. We have now generated the Ramachandran plot by plotting the dihedral angles of the residue between the two interacting carbonyl groups. The plot indicates that the reciprocal interactions are mainly concentrated in the polyproline

II (PPII), β -turn and right-twisted β -strand regions (See **RRC Fig. 13a** below). Unlike the one-sided $n \rightarrow \pi^*$ interactions reported previously that are abundant in proteins, the abundance of these newly discovered reciprocal $C=O \cdots C=O$ interactions is low ($\sim 7.2\%$). Secondary structure analyses using Stride show that reciprocal $C=O \cdots C=O$ interactions have considerable abundance in random coils ($\sim 20\%$) and turn regions (10%) of proteins but negligible presence in α -helices. This is in contrast to the one-sided $n \rightarrow \pi^*$ interactions that are most abundant in α -helices. As PPII helix is not included as an independent secondary structure in most secondary structure prediction programs, many PPII helices remain unassigned even though they are present in the experimentally solved structures. We observed that the coil regions having reciprocal $C=O \cdots C=O$ interactions are dominated by PPII structures. We have confirmed this by plotting the ϕ , ψ angles of residues in the random coil regions having reciprocal interactions (See **RRC Fig. 13b** below). This is not surprising given that Stride program fails to predict PPII regions and PPII conformations are known to dominate coil regions of folded proteins (Chem. Rev. **2006**, 106, 1877-1897). These results are now discussed in the revised manuscript and included in Figure 4a-b and Table 4 of the revised manuscript.

RRC Fig. 13. a, Ramachandran plot generated by plotting torsion angles (ϕ , ψ) of all residues in 2184 protein structures (blue) and torsion angles (ϕ , ψ) of the residue between the two interacting carbonyl groups involved in reciprocal $C=O \cdots C=O$ interactions (yellow). **b**, Ramachandran plot generated by

plotting torsion angles (ϕ , ψ) of all residues in 2184 protein structures (blue) and torsion angles (ϕ , ψ) of the residue between the two interacting carbonyl groups involved in reciprocal $C=O \cdots C=O$ interactions present only in the coil regions (yellow).

Reviewer's comment: With regard to the frequency of these interactions in different secondary structure types or different amino acids, it is important that the data be normalized. For example, in Figure 4a, could it be that the interactions are observed more frequently in “coil” than in “turn” because “coil” is simply more frequent than “turn”? The data from the last column of Extended figure 5a would be more appropriate. Similarly, the plots in Figures 4b and 4c should be normalized to the frequency of the amino acids. In particular, the observation that Leu-Pro is most common might simply be due to the fact that leucine is the single most common amino acid in proteins. When normalized to account for amino acid frequency, the data might show that other residues have stronger preferences for these reciprocal interactions. This is also relevant for Figure 4d, as hydrolases are the most well-represented class of enzymes in the PDB, and therefore might not be more prone to reciprocal $n \rightarrow \pi^*$ interactions than other classes. Finally, it is unclear what “frequency” is considered in Extended Figure 5b. Are there really proteins where 92% of residues engage in reciprocal $n \rightarrow \pi^*$ interactions? Perhaps “frequency” in this last case actually refers to the number of examples in that protein?

Our Response: We agree with the reviewer that normalized data/plots would provide better insights. We have now provided normalized data/plots for secondary structure distribution, amino acid distribution and amino acid pair distribution instead of absolute numbers.

- 1. Secondary structure (SS) distribution plot:** Percentage of amino acids involved in reciprocal interactions is defined as:

(Number of amino acids in the SS involved in reciprocal $n \rightarrow \pi^*$ interaction x 100) ÷ (Total number of amino acids in that SS)

RRC Fig. 14. Plot of percentage of amino acids involved in reciprocal interactions in various secondary structures.

- To find out % distribution of each amino acid to participate in reciprocal $n \rightarrow \pi^*$ interaction, we used the following formula

(Number of the particular amino acid in reciprocal $n \rightarrow \pi^*$ interaction x 100) ÷ (Total number of that particular amino acid in the proteins studied here)

RRC Fig. 15. Plot showing percentage distribution of amino acids involved in reciprocal $C=O \cdots C=O$ interactions.

3. To find out the % distribution of amino acid pairs to participate in reciprocal $n \rightarrow \pi^*$ interactions, we used the following formula.

(Number of a particular amino acid pair in the reciprocal $n \rightarrow \pi^*$ interaction x 100) ÷ (Total number of that amino acid pair in the proteins studied here)

RRC Fig. 16. Plot showing percentage distribution of amino acid pairs involved in reciprocal $C=O \cdots C=O$ interactions.

The changes that have occurred to our results due to normalization is discussed below.

1. In case of the abundance of reciprocal interactions in secondary structures, the trend is almost similar before and after normalization except that reciprocal interactions are found more in 3_{10} -helices compared to α -helices. α -helices are predominant in most proteins but only a handful of α -helix residues are involved in reciprocal interaction. The abundance of reciprocal interactions in various secondary structures is now presented in Table 4 of the revised manuscript.
2. After normalization, the amino acid percentage distribution differs from the frequency distribution provided earlier although proline remains the most abundant amino acid to participate in reciprocal $n \rightarrow \pi^*$ interaction. However, normalization has increased the percentage distribution of proline residues (24%). These results are presented in Figure 4c of the revised manuscript.
3. The plot of the amino acid pair that participate in reciprocal $n \rightarrow \pi^*$ interactions after normalization has changed significantly from the one before normalization. After normalization, the Pro-Pro residue pair is the most predominant one to partake in the reciprocal $n \rightarrow \pi^*$ interaction. These results are presented in Figure 4d of the revised manuscript.

We have now incorporated these changes in the revised manuscript. We have now removed plot 4d as this seems not very informative.

In extended figure 5b, by frequency we mean the number of instances of reciprocal $n \rightarrow \pi^*$ interaction in the particular protein (PDB code: 4LGY). In case of 4LGY, 92 instances of reciprocal

$n \rightarrow \pi^*$ interaction exist, whilst only 7 % of the amino acid residues present in the protein partake reciprocal $n \rightarrow \pi^*$ interaction. A separate column of the percentage of residues involved in reciprocal interaction is now incorporated in Supplementary Table 9 to clear any confusion.

Reviewer's comment: It is unclear where the data from Extended Figure 6 come from. Which structures were subjected to these calculations? Importantly, panel (a) appears to concern different molecules than panels (b/c), which are still different from those of panels (d/e). The datasets for these panels should be made clear. In addition, what is the horizontal axis for Extended Figures 6f/g? What conclusions are to be drawn from these data?

Our response: We have now brought this Figure to the main manuscript in the revised version (Figure 6). This Figure is necessary to understand the nature of reciprocal $C=O \cdots C=O$ interactions. The nature of $C=O \cdots C=O$ interactions is highly debated in the literature. While some consider them $n \rightarrow \pi^*$ orbital interaction, others believe them to be dipolar in nature. We have discussed reciprocal $C=O \cdots C=O$ interactions as orbital interactions ($n \rightarrow \pi^*$ and $\pi \rightarrow \pi^*$) because of the following reasons. Firstly, the plots of the $n \rightarrow \pi^*$ and sum of $n \rightarrow \pi^*$ and $\pi \rightarrow \pi^*$ orbital interaction energies against the $O \cdots C$ distances (d) show a strong correlation [See Figure below] (Figure 6 in the revised manuscript). **In RRC Fig. 17a**, we have plotted the distances (d_1 and d_2 values) against the $n \rightarrow \pi^*$ interaction stabilization energies (NBO second order perturbation energies $E^1_{(n \rightarrow \pi^*)}$ and $E^2_{(n \rightarrow \pi^*)}$) reported in Table 1-3. The plot suggest that the stabilization energies $E_{(n \rightarrow \pi^*)}$ for $n \rightarrow \pi^*$ interactions decreases with an increase in the $O \cdots C$ (d) in synthetic molecules **1-8**, molecules taken from CSD and interacting amino acid pairs obtained from PDB (Table 1-3). The plot of overall orbital interaction energies [sum of $n \rightarrow \pi^*$ $\{E_{(n \rightarrow \pi^*)}\}$ and $\pi \rightarrow \pi^*$ $\{E_{(\pi \rightarrow \pi^*)}\}$] interaction energies reported in Table 1-3 also shows a similar correlation with $O \cdots C$ (d) distances (**RRC Fig. 17b**) (Fig. 6b in the

revised manuscript). This correlation indicate that orbital interaction is the major mechanism for the stabilization of these reciprocal $C=O \cdots C=O$ short contacts. Secondly, $C=O \cdots C=O$ torsion angles of the carbonyl groups involved in reciprocal interactions indicate a net zero dipole-dipole interaction eliminating the possibility of these interactions being dipolar in nature. To emphasize this point, in **RRC Fig. 17c-d** (Figure 6c-d in the revised manuscript), we have plotted the values of $C=O \cdots C=O$ torsion angles of the 1432 molecules obtained from the CSD search. The torsion angle (T) between two dipoles could be used to understand the dipolar nature of interaction between them. As we know, antiparallel (T ~ 180°) dipoles attract and parallel dipoles (T ~ 0°) repel each other whereas two orthogonal dipoles (T ~ 90°) have net zero dipolar interaction. In case of reciprocal interaction, the $C=O \cdots C=O$ torsion angles show an orientational preference [$C=O \cdots C=O$ torsion angle falls in 60° to 90° or -60° to -90° range] as a consequence of the simultaneous restrictions on d_1 and d_2 (≤ 3.2 Å). However, the values of the $C=O \cdots C=O$ torsion angles (~90°) suggest that there would be almost net zero interaction between the dipoles, eliminating the possibility of strong dipolar interactions. Therefore, we conclude that orbital delocalization is the major driving force for the stabilization of reciprocal $C=O \cdots C=O$ interactions. However, an elaborate energy decomposition analysis may be required for the accurate deconvolution of various forces contributing to the stabilization of reciprocal $C=O \cdots C=O$ short contacts.

We have now omitted panel b-e as they have not provided much additional insight into understanding the nature of reciprocal $C=O \cdots C=O$ interaction.

RRC Fig. 17 | **a**, Plot of $n \rightarrow \pi^*$ interaction energies between the interacting carbonyl pairs against crystallographic O...C distances (d_1 and d_2) in molecules shown in Table 1-3. When the x-axis is d_1 , $E^1_{(n \rightarrow \pi^*)}$ is plotted in the y-axis and when the x-axis is d_2 , $E^2_{(n \rightarrow \pi^*)}$ is plotted in the y-axis. The d_1 , d_2 , $E^1_{(n \rightarrow \pi^*)}$ and $E^2_{(n \rightarrow \pi^*)}$ values are taken from Table 1-3. The $n \rightarrow \pi^*$ interaction energies were computed at B3LYP/6-311+G(2d,p) level of theory. The solid curve is drawn for convenience. **b**, Plot of overall orbital interaction energy (sum of $n \rightarrow \pi^*$ and $\pi \rightarrow \pi^*$ interaction energies) between the interacting carbonyl pairs against crystallographic O...C distances (d_1 and d_2) in molecules shown in Table 1-3. When the x-axis is d_1 , $E^1_{(n \rightarrow \pi^*)} + E^1_{(\pi \rightarrow \pi^*)}$ is plotted in the y-axis and when the x-axis is d_2 , $E^2_{(n \rightarrow \pi^*)} + E^2_{(\pi \rightarrow \pi^*)}$ is plotted in the y-axis. d_1 , d_2 , $E^1_{(n \rightarrow \pi^*)}$ and $E^2_{(n \rightarrow \pi^*)}$, values are taken from Table 1-3. $E^1_{(\pi \rightarrow \pi^*)}$

and $E^2_{(\pi \rightarrow \pi^*)}$ values are taken from Supplementary Table 3, Supplementary Table 6 and Supplementary Table 8. The orbital interaction energies were computed at B3LYP/6-311+G(2d,p) level of theory. The solid curve is drawn for convenience. **c**, Histogram plot showing the frequency of the $C^1=O^2 \cdots C^5=O^6$ dihedral angles (see Supplementary Fig. 3 for atom numbers) for 1432 molecules obtained from the CSD search. **d**, Histogram plot showing the frequency of the $C^5=O^6 \cdots C^1=O^2$ dihedral angles (see Supplementary Fig. 3 for atom numbers) for 1432 molecules obtained from the CSD search.

Reviewer's Comments

In addition to the important issues above, I believe that the manuscript would benefit from the following considerations.

(1) How is the fit of Extended Figure 6a determined? What is the resulting model?

Our response: The data points in Figure 6a is not fitted to any equation. The solid curve is used only for convenience to illustrate the trend.

2) I think that it is important to note that many of the CSD hits, at least as described by Extended Figure 3c, are very conformationally constrained, either by additional rings, highly substituted centers, or stereoelectronic effects. This makes it all the more remarkable that reciprocal $n \rightarrow \pi^*$ interactions are observed in the much less-constrained protein backbone.

Our response: The Figure pointed out by the reviewer only has 15 of 1432 molecules. In fact, there are many molecules that are less constrained than what was presented in that Figure. To get some insights into the structures of the small molecules having reciprocal $C=O \cdots C=O$ interactions, we have now manually analyzed small molecules from the CSD having 1,5-type reciprocal interactions

with both d_1 and $d_2 \leq 3.00 \text{ \AA}$. A total of 249 molecules fulfill the above criteria [1, 5-interaction; both d_1 & $d_2 \leq 3.00 \text{ \AA}$]. As can be anticipated, the nature of the two atoms/groups between the interacting carbonyl groups plays a key role to make the two carbonyl groups non coplanar and provides them the conformation required for reciprocal interactions (See **RRC Table 5** below). These results are now included in Supplementary Table 7. Interestingly, majority of these molecules (117, ~47%) have one heteroatom and one chiral carbon between the two interacting carbonyl pairs, a feature that resembles peptides and proteins.

RRC Table 5. Analysis of 1, 5-reciprocal interactions from the CSD with both d_1 and $d_2 \leq 3.00 \text{ \AA}$. A total of 249 molecules were obtained that have 1, 5-reciprocal $\text{C}=\text{O}\cdots\text{C}=\text{O}$ interactions with both d_1 and $d_2 \leq 3.00 \text{ \AA}$.

Groups Between the two C=O group	Number of molecules	Percentage (%)
One heteroatom (O or N), one chiral C atom	117	46.98
Both are chiral C atoms	61	24.49
One heteroatom (O or N), one achiral C atom	28	11.24
Both sp^2 C atoms [the C=O groups are attached to 1,2-positions of benzene, naphthalene or non-aromatic rings]	21	8.43
Both are heteroatoms (O or N)	14	5.62
One chiral and one achiral C atoms	4	1.60
Both are achiral C atoms [attached to rings with conformational constraints]	4	1.60

3) A ChemDraw figure would be helpful for interpreting Figure 2e/f.

Our response: A ChemDraw Figure is included now in the Figure 2e of the revised manuscript.

4) The authors have not commented on the types of small molecules that form these reciprocal $n \rightarrow \pi^*$ interactions, which might reveal some interesting trends.

Our response: To get some insights into the structures of the small molecules having reciprocal $C=O \cdots C=O$ interactions, we manually analyzed small molecules from the CSD having 1,5-type reciprocal interactions with both d_1 and $d_2 \leq 3.00 \text{ \AA}$. A total of 249 molecules fulfill the above criteria [1, 5-interaction; both d_1 & $d_2 \leq 3.00 \text{ \AA}$]. As can be anticipated, the nature of the two atoms/groups between the interacting carbonyl groups plays a key role to make the two carbonyl groups non coplanar and provides them the conformation required for reciprocal interactions (See **RRC Table 5** above). These results are now included in Supplementary Table 7. Interestingly, majority of these molecules (117, ~47%) have one heteroatom and one chiral carbon between the two interacting carbonyl pairs, a feature that resembles peptides and proteins. Analysis of the substituents attached to the carbonyl groups show that of 249 molecules, only 24 (9.6%) have the presence of strong electron withdrawing groups attached to at least one $C=O$ group.

5) It would be helpful for the authors to show illustrative examples of the interactions they find in proteins, particularly in different secondary structure contexts. For example, the authors note that “ α -helices that have two-sided $n \rightarrow \pi^*$ interactions are distorted, while the β -sheets having two sided $n \rightarrow \pi^*$ interactions are twisted.” It would be very helpful for readers to be able to visualize these distortions, as they could have important consequences for protein structure. Such examples would, in my opinion, be more informative than the examples shown in Extended Figure 4.

Our response: We agree with the reviewer. We have now included a Figure in the revised manuscript (Fig. 5 in the revised manuscript) to illustrate the reciprocal interactions found in different secondary structures. Please see the **RRC Fig. 18** below.

RRC Fig. 18. Reciprocal $C=O \cdots C=O$ interactions present in various secondary structure is shown. **a**, PPII- helix. **b**, β -turn. **c**, Right-twisted β -strand. **d**, α -helix. **e**, interface of α -helix and β -sheet.

Reviewer #1 (Remarks to the Author):

The authors have responded to my comments very well, and I feel that the data presentation, arguments, and the overall paper are much improved. In my view, the revised paper is now suitable for publication in Nature Communications.

Reviewer #2 (Remarks to the Author):

The authors have appropriately responded to most of my points BUT my main initial concern has not been clarified.

My main point of criticism was as follows "... the authors need to demonstrate – at least for their sequence of hydrazine derivatives in Fig. 1c – that geometry optimization of the remaining degrees of freedom (bond lengths, angles, and dihedrals) has a negligible effect on the overlap of the computed n/π^* orbitals and on the NBO energies in comparison to calculations done on the unrelaxed X-ray coordinates."

The authors replied "We agree with the reviewer that providing optimized geometry and electronic energy calculation on optimized geometry is the common practice."

Then, they write that they have performed new unconstrained calculations of a single molecule (compound 6). However, they observed that the molecular geometry of the chloride atom distorted significantly from its starting (X-ray) geometry - which is of course UNWANTED.

They argue that "one would have to come up with the best theoretical method and basis set to study these interaction, which itself would be a separate project for investigation."

IF this means that the level of theory used here is NOT ADEQUATE to properly describe the electronic structure of the molecules considered here, in my view this also casts doubts on the reported interactions between orbitals of these molecules.

Hartree-Fock and DFT calculations are pretty cheap methods these days.

The authors close their discussion of the calculations on compound 6 with "As this study deals with the possibility of reciprocal interactions in the crystal geometries of small molecules and proteins, to avoid such deviation from crystal geometries, we avoided free optimization the molecules. "

Contrary to this statement, the title of the manuscript and everything else suggest that the authors believe that reciprocal interactions are important for small molecules and proteins in general, and not only when they adopt their crystal geometries and are in the packing environment of a crystal.

Reviewer #3 (Remarks to the Author):

The revisions submitted for the manuscript regarding "Reciprocal carbonyl-carbonyl interactions in small molecules and proteins" resolve most of the key issues raised during initial review. The authors are to be commended for the thoroughness of their analysis. However, a few issues remain to be addressed:

1) I do not believe that the set of protein structures examined in this manuscript is of sufficient resolution to report data regarding carbonyl pyramidalization, which is a most subtle distortion. Previous studies of this parameter in proteins have focused on ultrahigh-resolution structures (less

than 1.0 Å), which probably represent only about 10% of the structures considered herein. Therefore, I am not sure that these data are reliable, and I would discourage their inclusion (Table 3) unless a more appropriate dataset is examined.

2) I still do not believe that the diacyl hydrazines can be used to demonstrate cooperativity of $n \rightarrow \pi^*$ interactions because the two interaction distances are not independent from one another. When carbonyl 1 gets closer to carbonyl 2, it is necessary that carbonyl 2 gets closer to carbonyl 1. That is why there is such a strong correlation in Figure 1g. Cooperativity need not be present for this trend to be observed. In order to truly demonstrate cooperativity, a change in the strength of one interaction must affect the strength of an independent interaction. In other words, as the interaction between carbonyls 1 and 2 gets stronger, the interaction between carbonyls 2 and 3 should also get stronger. Without that type of experimental design, I do not believe that cooperativity can be invoked.

3) I apologize for the poor phrasing of my question regarding the overlap between the datasets of this manuscript and that of Bartlett et al. The absolute number of interactions identified by the two studies cannot be compared, as the authors note, due to differences in the size of the protein dataset under examination. However, my interest is not in the absolute number of interactions present in each dataset, but rather in the relative number that would be identified using the criteria of Bartlett et al. compared to the number that would be identified using the criteria employed in this manuscript. The important piece of information is that 6.5% of the interactions identified by this study would have been identified using the criteria reported by Bartlett et al., demonstrating that reciprocal carbonyl interactions are largely distinct from sequential $n \rightarrow \pi^*$ interactions reported previously. The discussion of the differences in the protein datasets is therefore unnecessary and can be removed.

Robert W. Newberry
robert.newberry@ucsf.edu

Reviewer #4 (Remarks to the Author):

The authors report studies of reciprocal carbonyl-carbonyl interactions in small molecules and proteins. This is a somewhat unexplored previously type of interactions and it can play a significant role in determining 3-D protein structure. The manuscript is well written and generally clear. The level of science is good. In my opinion, the article should be publishable once the following issues have been addressed:

(1) The authors call these interactions "unprecedented" in the introduction part of the paper. I do not believe the interactions themselves are without a precedent. Moreover, I would recommend to tone down this and similar sentences in general.

(2) More thorough extensive analysis should be done for the computational results. This should definitely include analyzing charge redistribution.

Reviewer #2

Reviewer's comment: The authors have appropriately responded to most of my points BUT my main initial concern has not been clarified.

My main point of criticism was as follows "... the authors need to demonstrate – at least for their sequence of hydrazine derivatives in Fig. 1c – that geometry optimization of the remaining degrees of freedom (bond lengths, angles, and dihedrals) has a negligible effect on the overlap of the computed n/π^* orbitals and on the NBO energies in comparison to calculations done on the unrelaxed X-ray coordinates."

The authors replied "We agree with the reviewer that providing optimized geometry and electronic energy calculation on optimized geometry is the common practice."

Then, they write that they have performed new unconstrained calculations of a single molecule (compound 6). However, they observed that the molecular geometry of the chloride atom distorted significantly from its starting (X-ray) geometry - which is of course UNWANTED.

They argue that "one would have to come up with the best theoretical method and basis set to study these interaction, which itself would be a separate project for investigation."

IF this means that the level of theory used here is NOT ADEQUATE to properly describe the electronic structure of the molecules considered here, in my view this also casts doubts on the reported interactions between orbitals of these molecules.

Hartree-Fock and DFT calculations are pretty cheap methods these days.

The authors close their discussion of the calculations on compound 6 with "As this study deals with the possibility of reciprocal interactions in the crystal geometries of small molecules and proteins, to avoid such deviation from crystal geometries, we avoided free optimization the molecules. "

Contrary to this statement, the title of the manuscript and everything else suggest that the authors believe that reciprocal interactions are important for small molecules and proteins in general, and not only when they adopt their crystal geometries and are in the packing environment of a crystal.

Our response: We thank the reviewer for the suggestions. Reporting $n\rightarrow\pi^*$ interaction energies by performing NBO analysis on high resolution crystal structures without geometry optimization is

known and B3LYP/6-311+G(2d,p) level of theory [the same method that we used in this manuscript] worked well for that purpose (ref. 16 in the manuscript). Based on the reviewer's suggestions, we have now carried out geometry optimizations in compounds **1-8** by freezing the dihedral angles of the side chains involved in reciprocal interactions to their X-ray values (see Figure below and Supplementary Figure 3 in the revised manuscript) to find out if free optimization of the remaining degrees of freedom (bond lengths, angles, and dihedrals) have any effect on the overlap of the computed $n \rightarrow \pi^*$ interactions in comparison to calculations done on the unrelaxed X-ray geometries. We observed that reciprocal $n \rightarrow \pi^*$ interactions were retained after geometry optimizations but they became slightly weaker than what were observed from the NBO calculations on the crystal geometries (Supplementary Table 5). We also observed that, during gas phase geometry optimization, in absence of any packing and intermolecular forces that are present in the X-ray geometries, the Cl or Br atoms attached to the methylene carbons in **4, 6-8** moved to an anti-periplanar geometry (*trans*) with respect to the oxygen atom of the nearby carbonyl group (CO-II). This is probably due to higher hyperconjugative delocalization between the halogen lone pairs and carbonyl π^* orbital in the anti-periplanar geometry that would provide more stability to the isolated gas phase molecule. Note that such elongation of carbonyl-carbonyl ($O \cdots C$) short contacts (weakening of $n \rightarrow \pi^*$ interactions) in gas phase optimized geometry relative to the X-ray geometries are well known (see ref. 9,13 and 14 in the manuscript).

RRC Figure 1. Representative molecular geometry to show the frozen dihedral angles during geometry optimization.

Reviewer #3

Reviewer's comment: I do not believe that the set of protein structures examined in this manuscript is of sufficient resolution to report data regarding carbonyl pyramidalization, which is a most subtle distortion. Previous studies of this parameter in proteins have focused on ultrahigh-resolution structures (less than 1.0 Å), which probably represent only about 10% of the structures considered herein. Therefore, I am not sure that these data are reliable, and I would discourage their inclusion (Table 3) unless a more appropriate dataset is examined.

Our response: We agree with the reviewer. Accordingly, we have now removed this section from the manuscript. We have also removed columns 8-11 from Table 3 in the revised manuscript.

Reviewer's comment: I still do not believe that the diacyl hydrazines can be used to demonstrate cooperativity of $n \rightarrow \pi^*$ interactions because the two interaction distances are not independent from one another. When carbonyl 1 gets closer to carbonyl 2, it is necessary that carbonyl 2 gets closer to carbonyl 1. That is why there is such a strong correlation in Figure 1g. Cooperativity need not be present for this trend to be observed. In order to truly demonstrate cooperativity, a change in the strength of one interaction must affect the strength of an independent interaction. In other words, as the interaction between carbonyls 1 and 2 gets stronger, the interaction between carbonyls 2 and 3 should also get stronger. Without that type of experimental design, I do not believe that cooperativity can be invoked.

Our response: “When carbonyl 1 gets closer to carbonyl 2, it is necessary that carbonyl 2 gets closer to carbonyl 1. That is why there is such a strong correlation in Figure 1g.”

This is not true in general. If this were true we should have observed a similar correlations in Figure 2a and 3a, which we have not. The correlations were observed only for compounds **1-8** in Fig 1g. In Figures 2a and 3a, the d_1 vs. d_2 plots are scattered and, in most cases, d_1 does not increase or decrease with an increase or decrease of d_2 , which suggests that shortening of one donor-acceptor distance does not necessarily lead to a contraction of the second, reciprocal donor-acceptor distance.

“In other words, as the interaction between carbonyls 1 and 2 gets stronger, the interaction between carbonyls 2 and 3 should also get stronger. Without that type of experimental design, I do not believe that cooperativity can be invoked.”

I think, we have a disagreement in the use of terminology “cooperative” here. May be we should have used “synergistic” instead of “cooperative”. A famous textbook example of synergistic bonding is metal carbonyl bonding. As we know, in case of metal carbonyls, higher σ donation from CO to metal increases backdonation from metal’s filled π orbital to empty π^* orbital of CO, which are termed as “synergistic”. In case of synergistic interaction, we don’t need two independent interactions (1 \rightarrow 2 and 2 \rightarrow 3) as evident from metal carbonyl’s synergistic σ -bonding and π -back bonding.

In view of the above discussion, we have now replaced “cooperative” with “synergistic” in the manuscript. We believe that there is a strong possibility that the correlation between the two $n\rightarrow\pi^*$ interactions in compounds **1-8** is synergistic. However, to ascertain whether the origin of this correlation between the reciprocal $n\rightarrow\pi^*$ interactions in **1-8** is a result of synergistic effects or merely a result of the particular geometries adopted by **1-8** needs further investigation. Therefore, we have now modified this discussion in the revised manuscript and restricted ourselves to discuss the possibility of having synergistic interaction between the carbonyl groups in **1-8**.

Reviewer’s comment: I apologize for the poor phrasing of my question regarding the overlap between the datasets of this manuscript and that of Bartlett et al. The absolute number of interactions identified by the two studies cannot be compared, as the authors note, due to differences in the size of the protein dataset under examination. However, my interest is not in the absolute number of interactions present in each dataset, but rather in the relative number that would be identified using the criteria of Bartlett et al. compared to the number that would be identified using the criteria employed in this manuscript. The important piece of information is that 6.5% of the interactions identified by this study would have been identified using the criteria reported by Bartlett et al., demonstrating that reciprocal carbonyl interactions are largely distinction from sequential $n\rightarrow\pi^*$ interactions reported previously. The discussion of the differences in the protein datasets is therefore unnecessary and can be removed.

Our response: We have now modified this section and removed the discussion on the differences in the protein datasets from the revised manuscript.

Reviewer #4

Reviewer's comment: The authors call these interactions "unprecedented" in the introduction part of the paper. I do not believe the interactions themselves are without a precedent. Moreover, I would recommend to tone down this and similar sentences in general.

Our response: We have now modified this section and replaced “unprecedented” with “unexplored”. Please see the revised manuscript.

Reviewer's comment: More thorough extensive analysis should be done for the computational results. This should definitely include analyzing charge redistribution.

Our response: To analyze charge redistribution, we first took help of NBO deletion analysis. Due to $n \rightarrow \pi^*$ electron delocalization between donor oxygen lone pair and acceptor carbonyl π^* orbital of C=O bond, there should be a reduction on the electronic charge on the donor oxygen atom and a corresponding increase in the electronic charge on the π^* orbital of the acceptor carbonyl group. Conversely, deletion of $n \rightarrow \pi^*$ interaction should increase charge on the donor oxygen atom's lone pair and decrease charge on the acceptor carbonyl's π^* orbital, which could be studied by using NBO deletion analysis. Therefore, NBO deletion analysis should provide some understanding of the increase or decrease of charge on donor oxygen lone pair and acceptor carbonyl's $\pi^*_{C=O}$ orbital. Importantly, this change in charges should correlate with the strength of $n \rightarrow \pi^*$ interactions and hence O...C distances. We have now carried out NBO deletion analysis (Supplementary Table 13 of revised manuscript) and observed that deletion of $n \rightarrow \pi^*$ interactions increases charge on donor oxygen lone pair (n_O) and depletes it on the acceptor carbonyl $\pi^*_{C=O}$ orbital. Although the values of charge increase or decrease is small, these values correlate well with O...C distances suggesting electron delocalization between the two carbonyl groups involved in C=O...C=O interactions (Supplementary Table 13 of revised manuscript). Similarly, deletion of $\pi \rightarrow \pi^*$ interactions between the carbonyl groups leads to charge increase in the donor π orbital and decrease in the acceptor π^* orbital, which can be correlated to the strength of C=O...C=O interactions. The overall accumulation of charges on the acceptor carbonyl $\pi^*_{C=O}$ orbitals due to donation from the oxygen lone pairs and $\pi_{C=O}$ orbital of donor carbonyl is shown in Figure 6c-d of revised manuscript, which correlate well with the strength of C=O...C=O short contacts.

We have also carried out CHelpG charge calculations on compounds **1-8**, the molecules obtained from CSD search and amino acid dimers listed in Table 1-3. However, it is difficult to

correlate the charges on C and O atoms with the strength of C=O...C=O interaction. As we did not get reasonable correlation between charge on C and O with the strength of O...C interactions, we have not included these results in the manuscript. Please refer to the Tables below for the CHelpG charges.

Table 1. Charge on C and O atoms of carbonyl groups in **1-8** obtained from CHelpG calculations at B3LYP/6-311+G(2d,p) level of theory. For atom numbering see Figure 3d.

Comp	Charge on C ¹	Charge on O ¹	Charge on C ²	Charge on O ²	Charge separation between C ¹ and O ¹	Charge separation between C ² and O ²
	(electron)	(electron)	(electron)	(electron)	(electron)	(electron)
1	0.683	-0.57	0.775	-0.611	1.253	1.386
2	0.674	-0.549	0.689	-0.592	1.223	1.281
3	0.684	-0.57	0.751	-0.601	1.254	1.352
4	0.616	-0.544	0.684	-0.56	1.160	1.244
5	0.537	-0.518	0.733	-0.521	1.055	1.254
6	0.674	-0.552	0.732	-0.509	1.226	1.241
7	0.542	-0.531	0.593	-0.478	1.073	1.071
8	0.572	-0.541	0.733	-0.511	1.113	1.244

Table 2. CHelpG charges on C and O atoms of carbonyl groups involved in reciprocal interactions in molecules obtained from CSD search. CHelpG calculations were done at B3LYP/6-311+G(2d,p) level of theory.

Ref. Code	d ₁	d ₂ (Å)	Charge on C ₁	Charge on O ₁	Charge on C ₂	Charge on O ₂	Charge separation on C ₁ =O ₁	Charge Separation on C ₂ =O ₂
	(Å)	(Å)	(e)	(e)	(e)	(e)	(e)	(e)
BECLAW	2.816	2.894	0.611	-0.550	0.920	-0.523	1.161	1.444
JUHQEK	2.836	2.839	0.806	-0.596	0.680	-0.523	1.402	1.203
LEBRER	2.861	2.856	0.790	-0.578	0.555	-0.523	1.368	1.078
WOCHIF	2.885	2.832	0.831	-0.548	0.831	-0.542	1.379	1.373
ZUKVUY	2.887	2.829	0.902	-0.592	0.629	-0.540	1.494	1.169
LUCHEY	2.959	2.987	0.659	-0.627	0.816	-0.569	1.287	1.385
AZULUD	2.979	3.009	0.370	-0.446	0.708	-0.540	0.816	1.248
GECYEU	2.992	2.972	0.543	-0.457	0.382	-0.476	1.000	0.858
ACBZO01	3.042	3.016	0.798	-0.528	0.644	-0.479	1.327	1.123
CIQNEW	3.044	3.086	0.710	-0.555	0.374	-0.452	1.265	0.826
MODYIO	3.136	3.102	0.724	-0.613	0.768	-0.576	1.336	1.344
LAGTIX	3.137	3.119	0.605	-0.532	0.810	-0.544	1.137	1.354
GAPDIK	3.164	3.107	0.495	-0.494	0.394	-0.450	0.989	0.844
SUDAXAS01	3.171	3.101	0.644	-0.545	0.727	-0.479	1.190	1.206
YEXQOH	3.191	3.118	0.871	-0.540	0.491	-0.560	1.411	1.051

Table 3. CHelpG charges on C and O atoms of carbonyl groups involved in reciprocal interactions in amino acid pairs taken from PDB. CHelpG calculations were done at B3LYP/6-311+G(2d,p) level of theory.

AA pair	pdb	stretch	d ₁	d ₂	Charge on C ₁	Charge on O ₁	Charge on C ₂	Charge on O ₂	Charge separation on C ₁ =O ₁	Charge Separation on C ₂ =O ₂
			(Å)	(Å)	(e)	(e)	(e)	(e)	(e)	(e)
Ala-Ala	3s5m	402-403	2.485	2.999	0.523	-0.574	0.434	-0.514	1.097	0.948
Ala-Pro	2xu9	264-265	2.568	2.852	0.710	-0.548	0.545	-0.535	1.262	1.080
Ile-Pro	2opc	135-136	2.675	2.768	0.566	-0.545	0.574	-0.472	1.111	1.046
Leu-Pro	2x5o	141-142	2.712	2.771	0.821	-0.628	0.585	-0.552	1.449	1.137
Cys-Pro	1gcy	251-252	2.815	2.834	0.259	-0.511	0.639	-0.466	0.770	1.105
Gln-Lys	3wcq	9-10	2.815	2.816	0.713	-0.591	0.635	-0.553	1.304	1.188
Leu-Tyr	3u26	101-102	2.826	2.893	0.991	-0.674	0.625	-0.573	1.665	1.198
Pro-Pro	3cx2	186-187	2.89	2.838	0.582	-0.611	0.62	-0.519	1.193	1.139
Leu-Pro	1a2p	20-21	2.904	3.007	0.604	-0.596	0.626	-0.577	1.200	1.203
Thr-Glu	4pdy	108-109	2.918	2.936	0.827	-0.643	0.588	-0.579	1.470	1.167
Phe-Pro	1n08	76-77	2.943	2.941	0.220	-0.564	0.723	-0.547	0.784	1.270
Ala-Phe	4y1w	139-140	2.947	2.973	0.878	-0.643	0.529	-0.566	1.521	1.095
Ser-Asp	3ry4	79-80	2.952	2.981	1.092	-0.689	0.364	-0.585	1.781	0.949
Leu-Pro	1g5a	379-380	2.956	2.978	0.581	-0.577	0.626	-0.488	1.158	1.114
Lys-Pro	1k3i	50-51	2.975	2.938	0.573	-0.515	0.579	-0.511	1.088	1.090
Ala-Asp	2bi8	95-96	2.985	2.973	0.629	-0.645	0.608	-0.567	1.274	1.175
Ile-Pro	1o7i	107-108	2.986	3.082	0.620	-0.626	0.616	-0.589	1.246	1.205
Thr-Thr	3uxf	349-350	2.987	2.975	0.902	-0.629	0.814	-0.565	1.531	1.379
Pro-Ser	4psc	32-33	2.998	2.946	1.010	-0.645	0.584	-0.493	1.655	1.077
Asp-Pro	2vzp	2-3	3.026	3.026	0.728	-0.647	0.577	-0.566	1.375	1.143
Thr-Pro	1fj2	3-4	3.035	3.015	0.712	-0.632	0.537	-0.520	1.344	1.057
Ala-Pro	1g12	103-104	3.108	3.152	0.395	-0.568	0.621	-0.474	0.963	1.095
Ile-Pro	1e2w	208-209	3.114	3.143	0.594	-0.582	0.671	-0.494	1.176	1.165
Val-Pro	1jnd	294-295	3.119	3.135	0.778	-0.653	0.680	-0.499	1.431	1.179
His-Ser	1b6a	331-332	3.128	3.108	0.593	-0.550	0.607	-0.553	1.143	1.160
Phe-Gly	1odv	28-29	3.129	3.169	0.817	-0.628	0.656	-0.576	1.445	1.232
Ala-Leu	1ikp	388-389	3.141	3.183	0.650	-0.625	0.632	-0.522	1.275	1.154
Glu-Pro	1eu1	623-624	3.144	3.194	0.443	-0.600	0.614	-0.466	1.043	1.080
Ala-Arg	1ejd	119-120	3.175	3.171	0.830	-0.647	0.632	-0.539	1.477	1.171
Leu-Pro	1eb6	110-111	3.176	3.175	0.457	-0.610	0.689	-0.545	1.067	1.234

Reviewer #3 (Remarks to the Author):

I am satisfied with the adjustments made in this round of revision and feel the paper can be published as written. Though I remain unconvinced that cooperativity, or "synergy," has been adequately demonstrated by the diacyl hydrazines, the results are described in appropriately speculative terms.

Robert W. Newberry
robert.newberry@ucsf.edu

Reviewer #4 (Remarks to the Author):

In my opinion, the authors have addressed the reviewers' comments in a satisfactory fashion, and the manuscripts should now be publishable.